

# Parallel I/O in FMS and MOM5

Rui Yang[1], Marshall Ward[1,2], Ben Evans[1]

[1]National Computational Infrastructure, the Australian National University, Canberra, ACT 0200, Australia
[2]now at Geophysics Fluid Dynamics Laboratory, National Oceanic & Atmospheric Administration, Princeton, NJ 08540-6649, USA

*Correspondence to*: Rui Yang (rui.yang@anu.edu.au)

**Abstract.** We present an implementation of parallel I/O in the Modular Ocean Model (MOM), a numerical ocean model used for climate forecasting, and determine its optimal performance over a range of tuning parameters. Our implementation uses the parallel API of the netCDF library, and we investigate the potential bottlenecks associated with the model configuration, netCDF implementation, the underpinning MPI-IO library/implementations and Lustre filesystem. We investigate the performance of a global 0.25° resolution model using 240 and 960 CPUs. The best performance is observed when we limit the number of contiguous I/O domains on each compute node and assign one MPI rank to aggregate and write the data from each node, while ensuring that all nodes participate in writing this data to our Lustre filesystem. These best performance configurations are then applied to a higher 0.1° resolution global model using 720 and 1440 CPUs, where we observe even greater performance improvements. In all cases, the tuned parallel I/O implementation achieves much faster write speeds relative to serial single-file I/O, with write speeds up to 60 times faster at higher resolutions. Under the constraints outlined above, we observe that the performance scales as the number of compute nodes and I/O aggregators are increased, ensuring the continued scalability of I/O-intensive MOM5 model runs that will be used in our next generation higher resolution simulations.

## 1 Introduction

Optimal performance of a computational science model requires efficient numerical methods that are facilitated by the computational resources of the HPC platform. For each calculation in the model, the operating system (OS) must provide sufficient access to the data so that the calculation can proceed without interruption. This is particularly true in highly parallelised models on HPC cluster systems, where the calculations are distributed across multiple compute nodes, and often with strong data dependencies between the individual processes. I/O operations represent such a bottleneck, where one must manage the access of potentially large datasets by many processes while also relying on the available interfaces, typically provided by a Linux operating system (OS) to a POSIX parallel (or cluster) filesystem such as Lustre and through to distributed storage arrays. A poorly designed model can be limited by the data speed of an individual disk, or a poorly configured kernel may lack a parallel filesystem that is able to distribute the data transfer across multiple disks.



Datasets in climate modelling at the highest practical resolutions are typically on the order of gigabytes in size per numerical field, and dozens of such fields may be required to define the state of the model. For example, a double precision floating point variable of an ocean model over a grid of approximately 0.1° horizontal resolution and 75 vertical levels will typically require over 5 GiB of memory per field. Over 20 such fields may be necessary to capture the model state and preserve

bitwise reproducibility, and the periodic storage of model output may involve a similar number of variables per diagnostic timestep. A typical one-year simulation can require reading hundreds of gigabytes of input data, and can produce terabytes of model output. For disk speeds of 350 MB/sec, a serial transfer of each terabyte would take approximately one hour and can severely burden the model runtime. For such models, efficient I/O parallelisation is a critical requirement, and the increase in future scalability requires further improvements in I/O efficiency. Parallel I/O can describe any skilful decomposition of the

reading and writing of data across multiple threads, processes, compute nodes, or physical storage. Many climate models, and ocean models in particular, can be characterised as hyperbolic PDE solvers, which are naturally decomposed into numerically solvable subdomains with only local data dependencies (Webb et al., 1996, 1997), and it is natural to consider parallel I/O operations which follow a similar decomposition.

In the simplest and most extreme case, the field is fully decomposed to match the computational decomposition of the

model, so that the data used by each process element (PE), such as an MPI rank or an OpenMP thread, is associated with a separate file, with one file per PE. An example decomposition is shown in Figure 1, where the numbered black squares denote the computational domain of each PE. I/O operations in this case are fully parallelised. But this can also require an increasing number of concurrent I/O operations, which can produce an abnormal load on the OS and its target filesystems when such a model is distributed over thousands of PEs (Shan et al., 2007). It can also result in datasets which are distributed

over thousands of files, which may require significant effort to either analyse or reconstruct into a single file.

At the other extreme, it is possible to associate the data of all PEs with a single file, denoted by the red border in Figure 1. One method for handling single file I/O is to allow all PEs to directly write to this file. Although POSIX I/O permits concurrent writes to a single file, it can often compound the issues raised in the previous case, where resource contentions in the filesystem must now be resolved alongside any contentions associated with the writing of the data itself. Such methods

are rarely scalable without considerable attention to the underlying resource management, and hence we do not consider this method in the paper.

A more typical approach for single file I/O is to assign a master PE which gathers data from all ranks, and then serially writes the data to the output file. While this approach avoids the issues of filesystem resourcing outlined above, it also requires either an expensive collective operation and the storage of the entire field into memory, or a separation of the work

into a sequence of multiple potentially expensive collectives and I/O writes. These two options represent the traditional trade-off of memory usage versus computational performance, and both are limited to serial I/O write speeds.

In order to balance the desire for parallel I/O performance while also limiting the number of required files, one can use a coarser decomposition of the grid which groups the local domains of several PEs into a larger "I/O domain". A representative I/O domain decomposition, with I/O domains delineated by the yellow borders, is shown in Figure 1. Within



each I/O domain, one PE is nominated to be responsible for the gathering and writing of data. This has the effect of reducing the number of I/O processes to the number of I/O domains, while still permitting some degree of scalability from the concurrent I/O. Several models and libraries provide implementations of I/O domains, including the model used in this study (Maisonnave et al.,2017; Dennis et al., 2012). A similar scheme for rearranging data from compute tasks to selective I/O tasks is proposed and implemented in the PIO (parallel I/O) library which can be regarded as an alternative implementation

of I/O domains (Edwards et al., 2019).

Because the I/O domain decomposition will produce fields that are fragmented across multiple files, this often requires some degree of pre-processing. For example, any model change which modifies the I/O domain layout, such as an increase of CPUs, will often require that any fragmented input fields be reconstructed as single files. A typical 0.25° global simulations can require approximately 30 minutes of post-processing time to reconstruct its fields as single files; for global 0.1°

simulations, this time can be on the order of several hours, often exceeding the runtime of the model which produced the output.

In short, there are four fundamental approaches to model I/O, each with its respective trade-offs, which are outlined in Table 1. The first three approaches are common when using a single file per process, although multiple problems can arise regardless of whether the I/O operation is single-threaded or distributed (Shan et al., 2007).

One solution, presented in this paper, is to use a parallel I/O library with sufficient access to the OS and its filesystem which can optimise performance around such limitations and provide efficient parallel I/O within a single file. For example, a library based on MPI-IO can use MPI message passing to coordinate data transfer across processes, and can reshape data transfers to optimally match the available bandwidth and number of physical disks provided by a parallel filesystem such as Lustre (Howison et al., 2010). This eliminates the need for writer PEs to allocate large amounts of memory, and also avoids

any unnecessary post-processing of fragmented datasets into single files, while also presenting the possibility of efficient, scalable I/O performance when writing to a parallel filesystem.

In this paper, we focus on a parallel I/O implementation for the Modular Ocean Model (MOM), the principal ocean model of the Geophysical Fluid Dynamics Laboratory (GFDL) (Griffies et al., 2012). As MOM and its Flexible Modelling System (FMS) provide an implementation of I/O domains, it is an ideal platform to assess the performance of these different

approaches in a realistic model simulation. For this study, we focus on the MOM5 release, although the work remains relevant to the more recent and dynamically distinct MOM6 model, which uses the same FMS framework.

We present a modified version of FMS which supports parallel I/O in MOM by using the parallel netCDF API, and we test two different netCDF implementations: the PnetCDF library (Li et al., 2003) and the pHDF5-based implementation of netCDF-4 (Unidata, 2015). When properly configured to accommodate the model grid and the underlying Lustre filesystem,

both libraries demonstrate significantly greater performance when compared to serial I/O, without the need to distribute the data across multiple I/O domains.

In order to achieve the satisfied parallel I/O performance, it is necessary to determine the optimal settings across the hierarchy of I/O stack, including the user code, high level I/O libraries, I/O middleware layer and parallel filesystem. There



is a large number of parameters at each layer of the I/O stack, and the right combination of parameters is highly dependent

on the application, HPC platform, problem size and concurrency. Designing and conducting the I/O tuning benchmark is the key task of this work. It is of particular relevance to MOM/FMS users bottlenecked by I/O performance. But given the ubiquity of I/O in the HPC domain, the findings will be of interest to most researchers and members of the general scientific community.

The paper is outlined as follows. We first describe the basic I/O implementation of the FMS library, and summarise our

changes required to support parallel I/O. The benchmark process and tuning results are described and presented in the following section. Finally, we verify the optimal I/O parameter values by applying them to an I/O-intensive MOM simulation at higher resolution.

## 2 Parallel I/O Implementations in FMS

The MOM source code, which is primarily devoted to numerical calculation, will rarely access any files directly and instead

relies on FMS functions devoted to specific I/O tasks, such as the saving of diagnostic variables or the reading of an existing input file. Generic operations for opening and reading of file data occur exclusively within the FMS library, and all I/O tasks in MOM can be regarded as FMS tasks.

Within FMS, all I/O operations over datasets are handled as parallel operations, and are accessed by using the `mpp` module, which manages the model's MPI operations across ranks. The API resembles most POSIX-based I/O interfaces, and the

most important operations are the `mpp_open`, `mpp_read`, `mpp_write` and `mpp_close` functions, which are outlined below.

Files are created or opened using the `mpp_open` function, which sets up the I/O control flags and identifies which ranks will participate in I/O activity. Each rank determines whether or not it is assigned as a master rank of its I/O domain and, if so, opens the file using either the netCDF `nf_create` or `nf_open` functions.

The `mpp_write` interface is used to write data to a file, and supports fields of different data types and numbers of dimensions. Non-distributed datasets are contiguous in memory and are typically saved on every PE, and such fields are directly passed to the `write_record` function, which uses the appropriate netCDF `nf_put_var` function to write its values to disk.

When used with distributed datasets, `mpp_write` must contend with both the accumulation of data across ranks and the

non-contiguity of the data itself, due to the values along the boundaries (or "halos") of the local PE domains which are determined by the neighbouring PEs. The `mpp_write` function supports the various I/O methods described in Section 1. For single-threaded I/O, the data on each PE must first strip its local halo data from the field and copy the interior values onto a local contiguous vector. These vectors are first gathered onto a single master rank, which passes the data to the `write_record` function. The alternative is to use I/O domains, where each rank sends its data to the master PE of its local





I/O domain in the same manner as the single-threaded method, but where each I/O domain writes to its own file. When using I/O domains, a postprocessing step may be required to reconstruct the domain output into a single file.

The `mpp_read` function is responsible for reading data from files and is very similar to `mpp_write` in most respects, including the handling of distributed data. In this function, `read_record` replaces the role of `write_record` and the netCDF `nf_get` functions replace the `nf_put` functions.

When I/O operations have been completed, `mpp_close` is called to close the file, which finalises the file for use by other applications. This is primarily a wrapper to the netCDF `nf_close` function.

The major code changes relevant to the parallel I/O implementations are outlined below.

- All implementations are fully integrated into FMS and are written in a way to take advantage of existing FMS functionality.

- netCDF files are now handled in parallel by invoking the `nc_create_par` and `nc_open_par` functions in the FMS file handler, `mpp_open`.

- All fields are opened with collective read/write operations, via the `NF_COLLECTIVE` tag. This is a requirement for accessing variables with unlimited time axis and also a necessary setting to achieve good I/O performance. When possible, the prefilling of variables is disabled to shorten the file initialization time.

- Infrastructure for configuring `MPI_Info` settings has been added to allow fine tuning of the I/O performance at the MPI-IO level.

- The root PEs of I/O domains, which we identify as I/O PEs, are grouped into a new communicator via FMS subroutines and used to access the shared files in parallel.

- The FMS subroutine `write_record` is modified to specify the correct start position and size of data blocks in the 150  I/O domain for each I/O PE.

- New FMS namelist statements have been introduced to enable parallel I/O support and features. An example namelist group is shown below.

```
&mpp_io_nml
 parallel_netcdf = .true. # enable parallel I/O (Default: .false.)
```
```
 parallel_read = .false.   # Enable parallel I/O for read operations (Default: .false.)
 pnetcdf = .false.         # enable PnetCDF backend (.true.) or HDF5 backend (Default: .false.)
 parallel_chunk = .true.   # Default (.false.) or customized chunk for netCDF-4 format (.true.).
 chunk_layout = cnk_x, cnk_y   # The user defined chunk layout if parallel_chunk is set as .true.
```





## 3 Parallel I/O Performance Benchmark

On large-scale platforms, I/O performance optimization relies on many factors at the architecture level (filesystem), software stack (high level I/O libraries), and the application (access patterns). Moreover, external noise from application interference and the OS can cause performance variability, which can mask the effect of an optimization.

Obtaining good parallel I/O performance on a diverse range of HPC platforms is a major challenge, in part because of complex interdependencies between I/O middleware and hardware. The parallel I/O software stack is comprised of multiple

layers to support multiple data abstractions and performance optimizations, such as the high-level I/O library, middleware layer, and a parallel filesystem (Lustre, GPFS, etc.). A high-level I/O library translates the application's data structures into a structured file format, such as netCDF-3 or netCDF-4. Specifically, PnetCDF and parallel HDF5 are the parallel interfaces to the netCDF-3 and netCDF-4 file formats, respectively, and they are built on top of MPI-IO. The middleware layer, which in our case is an MPI-IO implementation, handles the organization and access optimization from many concurrent processes.

The parallel filesystem handles any accesses to files stored on the storage hardware in data blocks.

While each layer exposes tunable parameters for improving performance, there is little guidance for application developers on how these parameters interact with each other and how they affect the overall I/O performance. To address this, we select combinations of tunable parameters at multiple I/O layers covering parallelization scales, application I/O layout, high-level I/O libraries, netCDF formats, data storage layouts, MPI-IO and the Lustre filesystem. Although there is a large space of

tunable parameters at all layers of the parallel I/O stack, many parameters interact with each other and only the leading ones need to be investigated.

### 3.1 I/O Parameter Space

With over twenty tunable parameters across the parallel I/O stack, it can become intractable to independently tune every parameter for a realistic ocean simulation. In order to simplify this process, we conduct a pre-selection process by executing

a stand-alone FMS I/O program (`test_mpp_io`) which tests most of the fundamental FMS I/O operations over a domain of a size comparable to the lower resolution MOM5 benchmarks. After running this simplified model over the complete range of I/O parameters, we found that most of the parameters had no measurable impact on performance, and were able to reduce the number of relevant parameters to the list shown in Table 3, which are summarised below.

- Application: As described in the introduction, the `io_layout` parameter is used to define the distribution of I/O

domains in FMS. In the original distributed I/O pattern, multiple PEs are grouped into a single I/O domain within which a root I/O PE collects data from the other PEs and writes them into a separate file. In our parallel I/O implementation, the I/O domain concept is preserved in that data is still gathered from each I/O domain onto its root PE. The main difference is that these I/O PEs now direct their data to the MPI-IO library, which controls how the data is gathered and written to a single shared file. Retaining the I/O domain structures allows the application to

reorganize data in memory prior to any I/O operations and enables more contiguous access to the file.





- High-level I/O library: In general, the data storage layout should match the application access patterns in order to achieve significant I/O performance gains. The data layout of netCDF-3 is contiguous, whereas netCDF-4 permits more generalised layouts using blocks of contiguous subdomains (or "chunks"). To simplify the I/O tuning, we use the default chunking layout of netCDF-4 files, so that we can focus on the impact of other I/O parameters. We consider the impact of chunking on performance in the high-resolution benchmark.

- MPI-IO: There are many parameters in the MPI-IO layer that could dramatically affect the I/O performance. MPI-IO distinguishes between two fundamental styles of I/O: independent and collective. We only consider collective I/O in this work as it is required for accessing netCDF variables with unlimited dimensions (typically the time axis). All configurable settings on independent functions are thus excluded. The collective I/O functions require process synchronization, which provides an MPI-IO implementation the opportunity to coordinate processes and rearrange the requests for better performance. For example, as the high performance portable implementation packaged in MPICH and OpenMPI, ROMIO has two key optimizations, data sieving and collective buffering, which have demonstrated significant performance improvements over uncoordinated I/O. However, even with these improvements, the shared file I/O performance is still far below the single-file-per-process approach. Part of the reason is that shared file I/O incurs higher overhead due to filesystem locking, which can never happen if a file is only accessed by a unique process. In order to reduce such overhead, it is necessary to tune the collective operations. By reorganizing the data access in memory, collective buffering assigns a subset of client PEs as I/O aggregators. These aggregators gather smaller, non-contiguous accesses into a larger, contiguous buffer, and then write the buffer to the filesystem (Liao et al., 2008). Both I/O aggregators and collective buffer size can be set through MPI info objects (Thakur et al., 1999). For example, the number of aggregators per node is controlled by the MPI-IO hint `cb_config_list` and the total number of aggregators is specified in `cb_nodes`. To simplify the benchmark configuration, we always set `cb_nodes` to the total number of PEs and leave `cb_config_list` to control the actual aggregator distribution over all nodes. The collective buffer size, `cb_buffer_size`, is the size of the intermediate buffer on an aggregator for collective I/O. We initially set the value to 64 kB in the lower resolution model, and then evaluate its impact on the I/O performance of the higher resolution model.

- Lustre Filesystem: The positioning of files on the disks can have a major impact on I/O performance. On the Lustre filesystem, this can be controlled by striping the file across different OSTs (Object Storage Target). The Lustre stripe count, `striping_factor`, specifies the number of OSTs over which a file is distributed, and the stripe size, `striping_unit`, specifies the number of bytes written to an OST before cycling to the next OST. As there is limit of 165 stripes for a shared file on our Lustre filesystem, we set a range of stripe counts up to 165 to align the number of nodes. The stripe size should generally match the data block size of I/O operations (Turner et al., 2017); we find that the stripe size had limited effects on the write performance and the default 1MiB gave satisfactory I/O performance in our pre-selection process.





## 3.2 Configurations

The parallel I/O performance benchmark configurations are set up as shown in Table 3.

- Project size: We run a suite of 1-day simulations of the 0.25° global MOM-SIS model for each of the I/O parameters in Table 3. We then apply these results to a 1-day simulation of 0.1° models and validate the parallel I/O performance benefits. Each simulation is initialised with prescribed temperature and salinity fields and is forced by prescribed surface fields. The compute domain is represented by the horizontal grid sizes of $1440 \times 1080$ and $3600$

$\times 2700$ for the 0.25° and 0.1° models, respectively. Both configurations use a common 50 level vertical grid. Model output consists of several restart files in double-precision format, and a diagnostic output file in single-precision format. In order to produce significant I/O loads for such a short run, diagnostic output is saved after every timestep. In the 0.25° configuration, the model writes 70 GB of data to the diagnostic file over 48 time steps with the 0.25° configuration, and writes 2.7 TB of data over 288 steps with the 0.1° configuration model. Multiple

independent runs are repeated, and the shortest time is shown for each case.

- Domain layout: Domain layout depends on the total number of PEs in use. Two distinct CPU configurations, 240 and 960 PEs, are considered for the 0.25° model. The domain layout is $16 \times 15$ for 240 PEs and $32 \times 30$ for 960 PEs. In 0.1° model, grids are disturbed over 720 and 1440 PEs with the domain layout of $48 \times 15$ and $48 \times 30$ respectively. PEs are equally assigned in node majority along *x* direction of domain layout.

- High level I/O libraries and netCDF formats: The netCDF library provides parallel access to netCDF-4 formatted files based on the HDF5 library, and netCDF-3 formatted files via the PnetCDF library. HDF5 maintains two version tracks, 1.8.*x* and 1.10.*x*, in order to maintain the file format compatibility and the enabling of new features, such as the collective metadata I/O or Virtual Datasets. We are interested in checking the I/O performance to access different formats via various libraries as listed in Table 3.

We rely on the FMS I/O timers to measure the time metrics on opening (`mpp_open`), reading (`mpp_read`), writing (`mpp_write`) and closing (`mpp_close`) files together with the total runtime. The metric time contains both I/O operations and communications for generation of restart and diagnostic files and it takes the maximum walltime among all PEs. We do not attempt to compensate for variability associated with the Lustre filesystem, such as network activity or file caching, and rely on the ensemble to identify such variability.

Experiments are carried out on the NCI Raijin supercomputing platform. Each compute node consists of 2 Intel Xeon (Sandy Bridge) E5-2670 processors with a nominal clock speed of 2.6 GHz and containing 8 cores, or 16 cores per compute node. Standard compute nodes have 64 GB of memory shared between the two processors. A Lustre filesystem having 40 OSSes (Object Storage Servers) and 360 OSTs is mounted as the working directory via 56 Gb FDR InfiniBand connections.





## 4 Benchmark Results

### 4.1 Single-Threaded Single-File I/O of the 0.25° Model

The single-threaded single-file pattern of MOM5 is chosen as the reference to compare its I/O time with the parallel I/O methods. As with parallel I/O, this method creates a single output file and no post-processing is required. The I/O operation times and total execution times for our target libraries and PE configurations are shown in Table 4.

We can see that all benchmarks are I/O intensive and they are driven by file initialization and writing operations. Specifically, writing 4D dataset into the diagnostic file takes about 85% of total elapsed time. All other times are notably shorter than `mpp_write`.

The time used in writing data into netCDF-4 formatted files is about 10% longer than creating netCDF-3 formatted files. This reflects the fact that in serial I/O, the root PE holding the global domain data tends to write the file contiguously and it matches the contiguous data layout of netCDF-3 better than the default block chunking layout of netCDF-4.

Most I/O operations excluding `mpp_read` take longer time when the number of PEs increases from 240 to 960, due to the higher overhead from resource contention, I/O locking and data communication. This indicates that I/O time of MOM5 does not scale with number of PEs in the single-threaded single-file I/O pattern.

### 4.2 Parallel I/O performance Tuning of the 0.25° Model

#### 4.2.1 I/O Layout

As outlined in the introduction, I/O layout specifies the topology of I/O domains to which the global domain is mapped. In our parallel I/O implementation, we adapt the I/O layouts in FMS to define subdomains of parallel I/O activity. Only the root PE of each I/O domain is involved in accessing the shared output file via MPI-IO. A skilful selection of I/O layout can help to control the contentions on opening and writing of files. I/O layout is not involved in reading input files; all PEs access the input files independently when reading the grid and initialization data.

In this section we explore how I/O layouts affects the I/O performance. For each I/O layout, we adjust the number of stripe count and aggregator to approach the shortest I/O time.

In the 240 PE benchmark, the domain PEs are distributed over a two-dimensional grid of 16 PEs in the $x$-direction and 15 PEs in the $y$-direction, denoted as 16×15. On our platform, this corresponds to 16 PEs per node over 15 nodes. The experimental I/O subdomain is similarly defined as $n_x \times n_y$, where $n_x$= 1, 2, 4, 8, 16 and $n_y$= 3, 5, 15. On our platform, which uses 16 CPUs per node, we can interpret $n_x$ as the number of I/O PEs per node and $n_y$ as the number of I/O nodes. A schematic diagram of 16×15 PE domains and 4×3 I/O domains in 240 PE benchmark is shown in Figure 2. For the 960 PE benchmark, the PE layout is 32×30, which utilizes 960 CPU cores over 60 nodes. The experimental I/O layout is set as the combination of $n_x$ = 1, 2, 4, 8, 16, 32 and $n_y$= 15, 30. Note that in the case of $n_x$ = 1, there are $n_y$ I/O nodes and 1 I/O rank per I/O node. For all other cases in the 960-PE benchmark, there are 2 $n_y$ I/O nodes and ½ $n_x$ I/O PEs per I/O node.



The time metrics associated with different I/O layouts by using 240 and 960 PEs are measured and compared. All benchmark results are classified based on its library/format and the I/O layout, and we report the shortest observed time in each category.

In all benchmarks, the elapsed times for writing files in netCDF-4 and netCDF classic formats are very similar, as both are produced by utilizing HDF5 1.10.2 library. We will thus report performance among 3 libraries i.e. HDF5 1.8.20, HDF5

1.10.2 and PnetCDF 1.9.0.

The `mpp_open` metric measures both the opening time of input files and the creation time of output files. Its runtime versus I/O layout at 240 and 960 PE benchmarks is shown in Figure 3. In all of the experiments, PnetCDF has shorter `mpp_open` time than HDF5 due to the simpler netCDF-3 file structure. Both runtime and variability are much less in 240 PEs than in 960 PEs, indicating higher filesystem contention as the number of PEs is increased.

The `mpp_read` metric measures the time of all PEs to read data from the input files. Its dependence on I/O layout is shown in Figure 4. As I/O layout is only applicable to write rather than read operations, `mpp_read` time should be unaffected by I/O layout, as demonstrated in the figure. We also observe no consistent difference in `mpp_read` due to the choice of I/O library. As with `mpp_open` time, the `mpp_read` time is much higher in 960 PEs than 240 PEs which we again attribute to the increased file locking times and OST contentions when using more PEs.

The majority of I/O time is due to `mpp_write`, which depends strongly on the choice of I/O layout, as shown in Figure 5. In the 240 PE benchmarks, the write time drops quickly as we increase the number of I/O nodes ($n_y$) and more gently as the number of I/O PEs per node ($n_x$) is increased. The 960 PE benchmarks show a similar trend to the 240 PE results. The shortest write time of the 960 PE benchmarks is less than that of 240 PE ones, which indicates that parallel write time demonstrates the same degree of scalability. All libraries present similar `mpp_write` trend over I/O layout, as they

approach the shortest `mpp_write` time with moderate number of PEs per node (i.e. 2 or 4 PEs/node).

The `mpp_close` metric measures the time to close files, which involves synchronizations across all I/O ranks. Its dependence on I/O layout is shown in Figure 6. We observe that there is a notable loss of performance in the HDF5 1.8.20 library, which is exacerbated as both the number of nodes and I/O PEs per node are increased. As we shall demonstrate in a later section, this can be attributed to issues related to contentions between MPI operations and the use of the

`MPI_File_set_size` function in a Lustre filesystem. This effect is mitigated, although still present, in the HDF5 1.10.2 library. In contrast to all HDF5 libraries, PnetCDF has negligible `mpp_close` time as there are fewer metadata operations in netCDF-3 than netCDF-4.

The total elapsed time versus I/O layout for all libraries are plotted in Figure 7. The HDF5 1.8.20 takes more time than HDF5 1.10.2 to produce the netCDF 4 files, due to longer `mpp_write` and `mpp_close` time. The shortest total time for

HDF5 1.10.2 and PnetCDF 1.9.0 happens at an I/O layout of 8✕15 (8 PEs/node) for 240-PE and 4✕30 (2 PEs/node) for 960-PE. Comparing it with all other time metrics as shown above, `mpp_write` dominates the total I/O time.





The impact of I/O layout on each I/O component time indicates that excessive parallelism can give rise to high I/O contention within the file server and can diminish I/O performance. We could thus set up the delegated I/O processes to reduce the contention that is also detailed in other work (Nisar et al., 2008). The best I/O performance is achieved by using a

moderate number of I/O PEs per node, such as 8 I/O PEs/node in the 240-PE or 2 I/O PEs/node in the 960-PE benchmark. Each I/O PE collects data from other PEs within the same I/O domain and forms more contiguous data blocks to be written to disk. In the next section, we use the best-performing I/O layouts, 8×15 for 240 PE and 4×30 for 960 PE, to explore the optimal settings of Lustre stripe count and MPI-IO aggregator.

### 4.2.2 Stripe Count and Aggregators

The Lustre stripe count and the number of MPI-IO aggregators can be set as MPI-IO hints when creating or opening a file, and are the two major MPI-IO parameters affecting I/O performance. The MPI-IO hint `striping_factor` controls the total number of stripe counts of a file; `cbnode` sets the total number of collective aggregators; and `cb_config_list` controls the distribution of aggregators over each node. In ROMIO, there are competing rules which can change the interpretation of these parameters. For example, the total number of aggregators must not exceed the stripe count; otherwise,

it will always be set to the stripe count. To simplify the parameter space, we adopt the actual number of aggregators (denoted as `real_aggr`) and stripe counts (denoted as `real_stp_cnt`) as the basic parameters in tuning the I/O performance.

### 240 PEs

The variations of each time metric versus the number of aggregators and stripe counts for each library are plotted in Figure 8 for the 240 PE experiments.

The `mpp_open` time does not depend strongly on the number of aggregators. PnetCDF spends less `mpp_open` time than all HDF5 libraries.

The `mpp_read` time increases as the number of aggregators and stripe counts are increased. Runtime is independent of library, as expected for a serial I/O operation.

The optimal `mpp_write` time is observed when the aggregator and stripe counts are set to 60. The overall `mpp_write`

times are quite comparable among all HDF5 libraries and they are slightly higher than PnetCDF, as observed in the I/O layout timings.

The `mpp_close` times of the HDF5-based libraries are independent of the number of aggregators, and increase slightly as the stripe count is increased. HDF5 v1.8.20 spends a much greater time in `mpp_close` than HDF5 1.10.2. The `mpp_close` time is negligible for PnetCDF and shows no measurable dependence on aggregator and stripe count.

The total runtime shows similar dependences on stripe count and aggregators with `mpp_write`. The performance trend across libraries remains consistent over I/O tuning parameters, with PnetCDF showing the best performances followed by HDF5 1.10.2 and HDF 1.8.20. The optimal parameters for read and write operations was observed when we set the number





of aggregators and stripe count to 15 or 30. This corresponds to one or two aggregators per node, with all 15 nodes contributing to I/O operations.

**960 PEs**

The variations of each time metric on the number of aggregator and stripe count in all library/format bindings are plotted in Figure 9 for 960-PE experiments.

The metrics for the 960-PE benchmarks show a similar trend to the 240-PE benchmarks. Both `mpp_open` and `mpp_read` times increase from 240-PE to 960-PE, in most cases by a factor of two, due to the higher contentions due to accessing the same files. Using the smallest number of aggregators, namely 60 aggregators or 1 aggregator per node, together with an equal number of stripes, gives the best performance for both `mpp_open` and `mpp_read` times. The `mpp_write` times are shorter than those of 240-PE. As in previous results, PnetCDF shows the best performance, while HDF5 1.10.2 outperforms HDF5 1.8.20. We observe that the best write performance occurs when the number of aggregators and stripe counts are set to 60, or 1 per node. Overall, the total time is reduced when using 960 PEs.

In both the 240-PE and 960-PE experiments, the best I/O performance occurs when the Lustre stripe count matches the number of aggregators. Using a larger stripe count may degrade the performance, since each aggregator process must communicate with many OSTs and must contend with reduced memory cache locality when the network buffer is multiplexed across many OSTs (Bartz et al., 2015; Dickens et al., 2008; Yu et al., 2007).

**4.2.3 I/O Implementation Profiling Analysis**

The above benchmark results show performance variances among different libraries and formats. In order to explore the source of differences in performance, we have developed an I/O profile to capture I/O function calls at multiple layers of the parallel I/O stack, including netCDF, MPI-IO and POSIX I/O, without requiring source code modifications. It provides a passive method for tracing events through the use of dynamic library preloading. It intercepts netCDF function calls issued by the application and reroutes them to the tracer, where the timestamp, library function name, target file name, and netCDF variable name along with function arguments are recorded. The original library function is then called after these details have been recorded. It is applied similarly at the MPI-IO and POSIX I/O layers. We have disabled profiling of HDF5 and PnetCDF libraries, as both are intermediate layers. Profiling overheads were measured to be negligible in comparison to the total I/O time.

We apply the I/O profiler described above to the 240-PE benchmark experiments, using the optimal I/O parameters from the previous analysis. The profiling results are shown in the call path flow charts below for each library. The accumulated maximum PE time is presented within each function node and above call path links. The number of I/O PEs involved in each call path is also given in the brackets. Call paths with trivial elapsed time have been omitted.





As shown in Fig. 10, `nc_close` is the most time consuming netCDF function in the benchmark of HDF5 1.8.20/netCDF-4. Two underlying MPI-IO functions, `MPI_File_write_at` and `MPI_File_set_size`, consume the majority of time

within `nc_close`. HDF5 metadata operations are comprised of many smaller writes, and the independent write function `MPI_File_write_at` from each PE may give rise to large overheads due to repeated use of system calls. It is a known issue that using `MPI_File_set_size` on a Lustre filesystem which uses the `ftruncate` system call, has an unfavourable interaction with the locking for the series of metadata communications which the HDF5 library makes during a file close (Howison et al., 2010). In practice, this leads to relatively long close times and prohibits I/O scalability.

Aside from the metadata operations, reading and writing netCDF variables are conducted collectively via `MPI_File_read_at_all` and `MPI_File_write_at_all` functions, which retain good I/O performance when processing non-contiguous data blocks.

In the HDF5 1.10.x track, collective I/O was introduced to improve the performance of metadata operations. Collective metadata I/O can improve performance by allowing the library to perform optimizations when reading the metadata, by

having one rank read the data and broadcasting it to all other ranks. It can improve metadata write performance through the construction of an MPI derived datatype that is then written collectively in a single call. The call path flow of tuned 240-PE benchmark with HDF5 1.10.2/netCDF-4 is shown in Fig. 11.

It shows that `nc_close` now invokes `MPI_File_write_at_all` instead of `MPI_File_write_at` in HDF5 1.10.2 spends less time than HDF5 1.8.20. Furthermore, HDF5 1.10.2 has been modified to avoid `MPI_File_set_size` calls

when possible by comparing the library's EOA (End of Allocation) with the filesystems EOF and skipping the `MPI_File_set_size` call if the two matches. As a result, HDF5 1.10.2 spends much less time on `nc_close` function than HDF5 1.8.20. Aside from the metadata operations, the general write performance of the `nc_put_vara_double` and `nc_put_var1_double` functions show similar performance in netCDF 1.10.2 and 1.8.20 when accessing netCDF-4 formatted files.

The call path flow of the tuned 240-PE benchmark with PnetCDF is shown in Fig. 12. Due to the simpler file structure of netCDF-3, the `nc_close` function spends a trivial amount of time in `MPI_Barrier` and `MPI_file_sync` rather than invoking expensive `MPI_File_set_size` function calls, which explains the much shorter `mpp_close` time in the benchmark experiments. In addition, the function `nc_put_vara_double` also spends less time than the HDF5 libraries, which implies that the access pattern matches the contiguous data layout of netCDF-3 performs in a better way than the

default block chunking layout of netCDF-4.

### 4.2.4 Load Balance

Load balance is another factor which may strongly affect I/O performance. In Fig. 13 we compare the time distribution over PEs in 3 layers of the major write call path between HDF5 1.10.2 and PnetCDF.





In the benchmark of the HDF5 1.10.2, both `nc_put_vara_double` and `MPI_File_write_at_all` functions are

called by 8 PEs per node, as configured in the I/O layout of 8✕15. The POSIX write function is invoked by 2 PEs per node, as configured by the MPI-IO aggregator configuration, `real_aggr=30`. All three functions show good load balance, as one would expect since all I/O PEs participate in the collective I/O operations. There are overheads in the `nc_put_vara_double` and `MPI_File_write_at_all` functions, but there is a larger time gap between `MPI_File_write_at_all` and the POSIX `write` call, which reflects the communication overhead among aggregators

and other PEs associated with collective buffering. A similar pattern also appears in the PnetCDF profile. Although HDF5 1.10.2 and PnetCDF spend a similar amount of time on POSIX `write` calls, the aggregation overheads are much higher in HDF5. This suggests again that the conventional contiguous storage layout in netCDF-3 matches the access pattern better than the default block chunking layout of netCDF-4.

### 4.2.5 Serial Read and Parallel Read

As indicated in the above benchmark experiments, the write performance is optimised by choosing an appropriate number of I/O PEs, aggregators and Lustre stripe count. In contrast to `mpp_write`, the `mpp_read` time grows from 240-PE to 960-PE benchmarks and can potentially become a major performance bottleneck for a large number of PEs. Since I/O layout is not employed in the parallel read process and the input files may use different formats and data layouts, there is no means to skilfully tune the parallel read performance.

As noted earlier, the serial `mpp_read` time is relatively small and stable in both 240-PE and 960-PE benchmarks. This motivates us to combine the original serial read with the parallel write in order to approach the best overall I/O performance. The 960-PE benchmarks with an I/O layout of 4✕30 and using serial read (denoted here as `sread`) and parallel write methods are shown for the HDF5 1.10.2 and PnetCDF libraries. The performance is compared with the parallel read benchmarks (denoted as `pread`) in Figure 14.

The `mpp_read` time is much shorter in the serial read benchmarks than the parallel reads and it remains fixed as stripe count is increased. The `mpp_open` times increase with stripe count, but are otherwise consistent across the four benchmarks shown. The serial read is unaffected by the write performance and file closing times. As a result, the net serial read time is overall shorter than parallel read times in both HDF5 1.10.2 and PnetCDF benchmarks.

### 5 I/O Performance validation of 0.1° Model

The tuning results from the 0.25° model suggests that the best parallel I/O performance could be achieved with the following settings:

- Parallel write with
    - Moderate number I/O PEs per node to access the file, as defined by I/O layout.





o    1 or 2 aggregators per node, as defined by MPI-IO hints.

440                  o    Stripe count matching the number of aggregators, as defined by MPI-IO hints.

•    Serial read on input files with the same stripe count as parallel write.

In this section we apply the above settings to the 0.1° model and measure their impact on I/O performance. As shown in previous results, the HDF5 1.8.20 library is overall slower than the HDF5 1.10.2 due to its higher metadata operation overheads, so we focus on the HDF5 1.10.2 and PnetCDF libraries.

The domain layouts of the 720-PE and 1440-PE runs are 48✕15 and 48✕30, respectively. We choose I/O layouts of 3✕15 and 3✕30 for 720 and 1440 PEs respectively so there is one I/O PE per node. The number of aggregators is also configured to one per node, and the stripe count is set to the total number of aggregators, i.e. 45 and 90 for the 720-PE and 1440-PE runs, respectively. For all benchmark experiments, we use serial independent reads and parallel writes. The measured time metrics in 720-PE and 1440-PE runs for the HDF5 1.10.2 and PnetCDF libraries are shown in Table 5. The timings of the

original single-threaded single file I/O (SIO) pattern in 720-PE and 1440-PE runs are also listed for comparison.

As shown in Table 5, the original serial I/O pattern requires a very long time (about 6 hours) to create a large diagnostic file (2.7 TB) and multiple restart files (75 GB) in 720-PE runs. The serial 1440-PE runs exceeded the platform job time limit of 5 hours and could not be completed, but the lack of scalability of serial I/O indicated by 0.25° model (Table 4) suggest that the total time would be comparable to the 720-PE runs. We noticed that the PnetCDF timings are 20% faster than the HDF5

times, as also observed in the 0.25° model benchmarks. Both libraries have similar non-I/O times at each level of PE count, which comprise less than 5% of total runtime, demonstrating that the benchmarks are I/O intensive and that different libraries have no impact on the computation time.

The value of `mpp_write` in parallel I/O are much shorter than the serial times. In the 720-PE runs, the parallel write time is about 30 to 36 times faster than the serial time in both the HDF5 and PnetCDF libraries. Such speedups are reasonable

relative to the 720-PE configuration, which uses 45 I/O PEs, aggregators and stripe counts. In the 1440-PE benchmark, which also doubles our number of I/O PEs, aggregators, and stripe counts to 90, the parallel `mpp_write` runtime was further reduced by a factor of two. We also observe that the non-I/O compute time of MOM from 720-PE to 1440-PE runs was reduced by a factor of two, complementing the enhanced I/O scalability of the parallel I/O configuration and maintaining the high overall parallel scalability of the model for I/O intensive calculations.

The PnetCDF library shows better write performance than HDF5 in both serial and parallel I/O, as well as a much shorter time in `mpp_close`. To investigate such performance diversity, we have conducted further tests on changing the data layout of HDF5/netCDF-4.

All HDF5 performance results used the default block chunking layout, where the chunk size is close to 4 MB with a roughly equal number of chunks along each axis. We repeated these tests by customizing the chunk layout while keeping all other

I/O parameters unchanged. The chunk layout, (`ckx, cky`), could be defined such that the global domain grids are divided into `ckx` and `cky` segments along the X and Y axes, respectively. The `mpp_write` times and total runtimes of the 720-PE



runs for chunking layouts spanning values of ckx∈{1, 2, 3, 4} and cky∈{1, 3, 5, 15} are plotted in Figure 15. The performance of the default chunking layout of HDF5 and PnetCDF are also shown in the figure as a reference point.

The chunk layout of (1, 1) defines the whole file as a single chunk. In this case, it occupies the same contiguous data layout
with PnetCDF. Not surprisingly, the mpp_write time of chunk layout (1, 1) is nearly the same as that of PnetCDF/netCDF-3 as shown in Figure 16. In fact, the mpp_write time changes only slightly across cky values when for ckx=1. On the other hand, changing ckx values for a fixed cky value give rise to a steeply increasing mpp_write time. Given the conventional contiguous storage layout of a 4d variable (t, z, y, x), the time dimension varies most slowly, z and y vary faster, and x varies fastest. This is also true within a chunk and increasing ckx will produce more non-contiguous
chunks than increasing cky. This means an I/O PE needs more I/O operations to write a contiguous memory data block across multiple chunks along the increasing ckx than cky, and thus write times rise accordingly as shown in Figure 14. An exception case is ckx=3 as it used similarly short write time with ckx=1. This is because it matches the number of x divisions of I/O layout (3, 15) and each I/O PE needs only 1 operation to write a line of data with the fixed y value. Instead, for ckx=2 or ckx=4, each I/O PE may use two or more write operations to write a line of y as it crosses multiple chunks.
This makes the write time much longer for ckx∈{2, 4} than ckx∈{1, 3}.

The mpp_close time is negligible in all tests. By reducing the total number of chunks and thus the metadata operations overheads, the mpp_close time can also be controlled with the reasonable chunk layout. The total time presents the similar trend with mpp_write along different chunk layouts as shown in Figure 15.

Choosing a good chunk layout depends strongly on the I/O layout settings. Using a single chunk in the netCDF-4 file is
unnecessary as it resembles the same data layout as the netCDF-3 format. Adopting an I/O layout as the chunk shape is sufficient for achieving optimal performance if our intention is to create netCDF-4 formatted output files and to utilize more advanced features, such as compression and filtering operations.

**Conclusions**

We have implemented parallel netCDF I/O into the FMS framework of the MOM5 ocean model, and presented results which
demonstrate the I/O performance gains relative to single-threaded single-file I/O. We present a procedure for tuning the relevant I/O parameters, which begins with identifying the I/O parameters that are sensitive to overall performance by using a light-weight benchmark program. We then systematically measure the impact of this reduced list of I/O parameters by running the MOM5 model at a lower (0.25°) resolution and determine the optimal values for these parameters. This is followed by a validation of the results in the higher (0.1°) resolution configuration.
Several rules for tuning the parameters across multiple layers of the I/O stack are established to maintain the contiguous access patterns and achieve the optimal I/O performance. At the user application layer, I/O domains were defined to retain more contiguous I/O access patterns by mapping the scattered grid data to a smaller number of I/O PEs. We achieve the best




performance when there is at least one I/O PE per node, and there can be additional benefits to using multiple I/O PEs per node, although an excessive number of I/O PEs per node can impede performance.

At the MPI and Lustre levels of the I/O stack, it was found that the number of aggregators used in collective MPI-I/O operations and the number of Lustre stripe counts needed to be consistently restricted to no more than 2 per node in order to facilitate contiguous access and reduce the number of contentions between PEs.

An I/O profiling tool has been developed to explore overall timings and load balance of individual functions across the I/O stack. It was determined that the MPI implementation of particular I/O operations in the HDF5 1.8.20 library used by
netCDF-4 caused significant overhead when accessing metadata, and that these issues were largely mitigated in HDF5 1.10.2. Additional profiling of the PnetCDF 1.9.0 library showed that it did not suffer from such overhead, due to the simpler structure of the netCDF-3 format.

High-resolution MOM5 benchmarks using the 0.1° configuration were able to confirm that the parallel I/O implementations can dramatically reduce the write time of diagnostic and restart files. Using parallel I/O enables the scaling of I/O operations
in pace with the compute time and improves the overall performance of MOM5, especially when running an I/O-intensive configuration resembling our benchmark. The parallel I/O implementation proposed in this paper provides an essential solution that removes any potential I/O bottlenecks in MOM5 at higher resolutions in the future.

An investigation of data compression is not a part of this work, as traditionally it can only be used in serial I/O. We note that the more recent version of HDF5, 1.10.2, introduced support for parallel compression, and it is expected that the netCDF
library will soon follow. As the I/O layout generally picks up 1 to 2 I/O PE per compute node, it may produce chunks which are too large (i.e., too small number of chunks) for efficient parallel compression. In this sense, the default chunk layout of netCDF4 should also be considered as it gains acceptable write performance and has suitable chunk sizes more suitable for parallel compression.

**Code availability**

The source code is available on GitHub at https://github.com/NOAA-GFDL/FMS/tree/with-parallel-netcdf

**Author contributions**

RY and MW developed the parallel I/O code contributions to FMS. RY carried out all model simulations, as well as
performance profiling and analysis. RY and MW wrote the initial draft of the article. All co-authors contributed to the final draft of the article. BE supervised the project.

**Competing interests**

The authors declare that they have no conflict of interest.



**Acknowledgement**

This work used supercomputing resources provided by National Computational Infrastructure (NCI), the Australian National University.

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



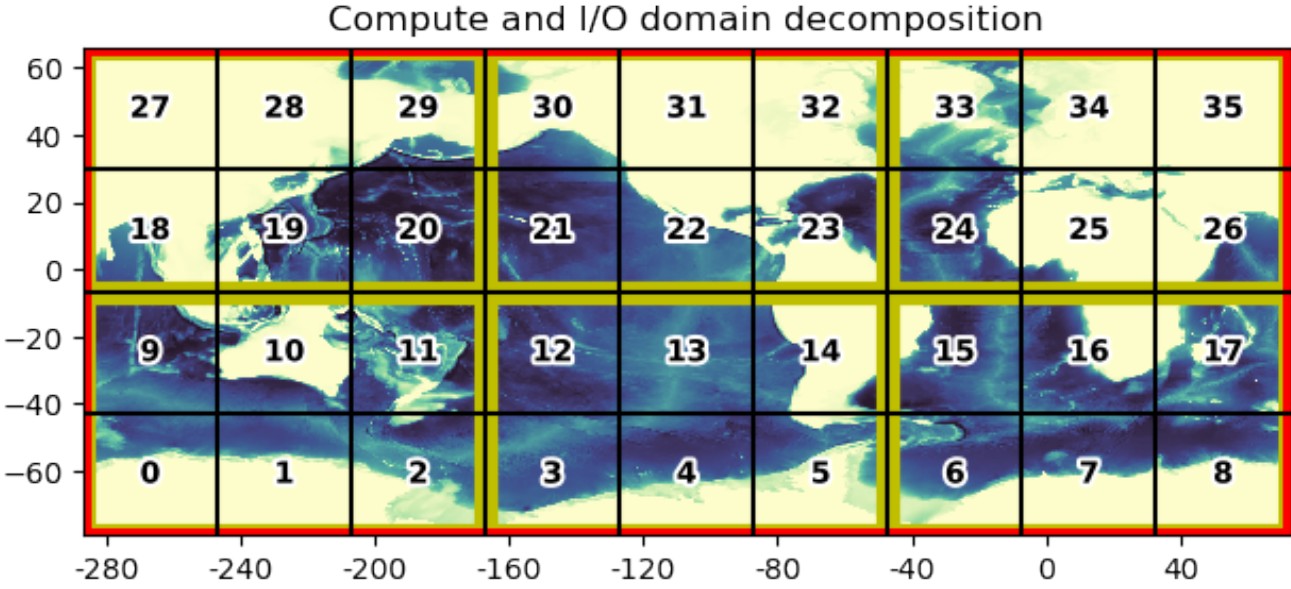

**Figure 1: A representative decomposition of a global domain. Black squares denote the computational domains of each process, and yellow boundaries denote the collection of computation domains into a larger I/O domain. The global domain is denoted by the red boundary.**

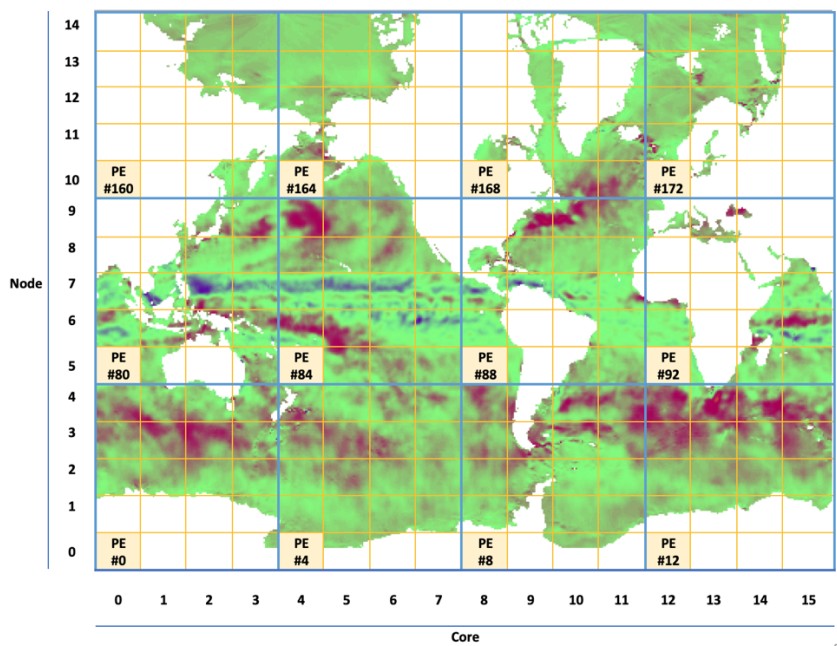

**Figure 2. A schematic diagram of 16×15 computation domain (☐) and 4×3 I/O domain (☐) with 12 I/O PEs (◯) in a 240 PE benchmark. The index of each I/O PE is labelled.**



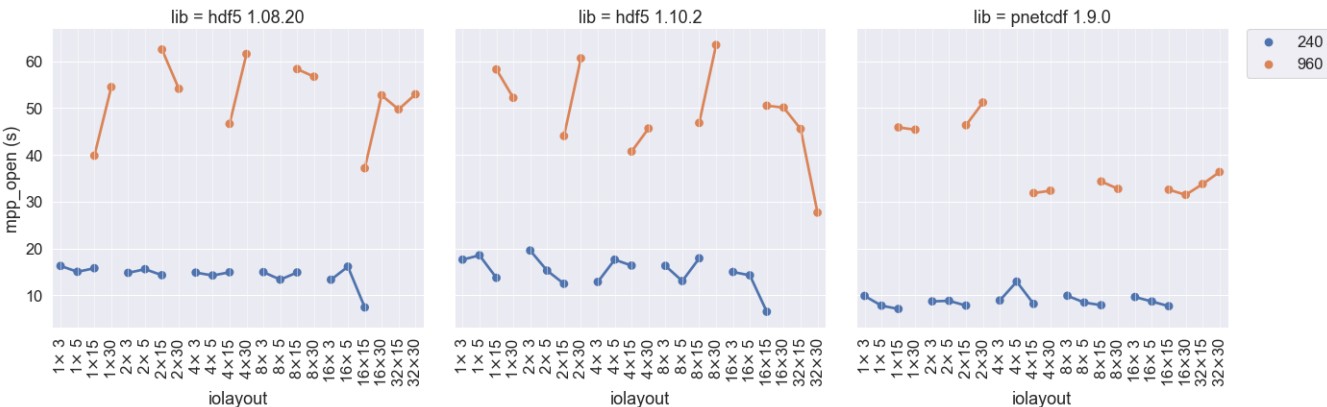

**Figure 3. `mpp_open` time (in sec.) versus I/O layout in different libraries and PE numbers. HDF5 times are generally larger than in PnetCDF, and the runtime increases as PEs increase from 240 to 960.**

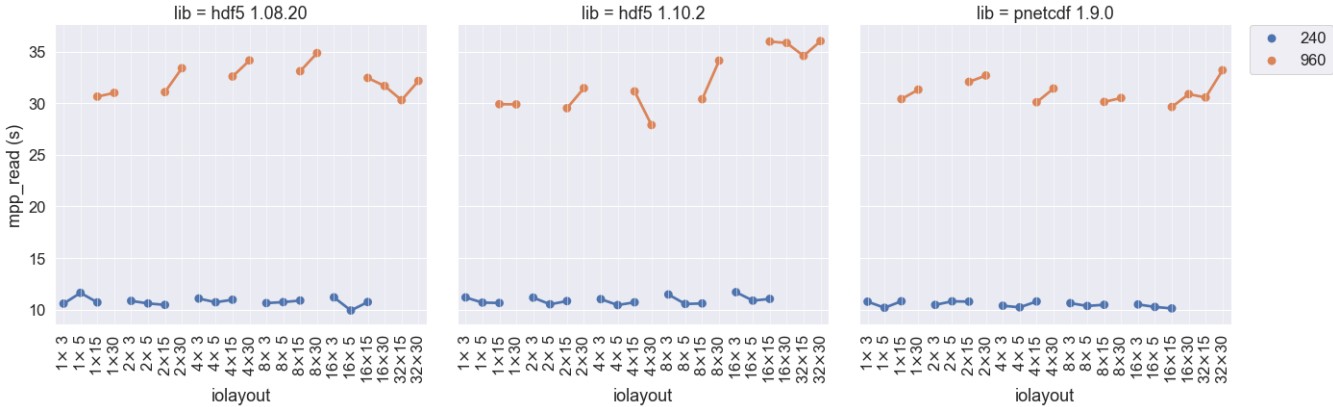

**Figure 4. `mpp_read` time (in sec.) versus I/O layout in different libraries and PE numbers. Read operations do not use I/O layout or parallel I/O, and runtimes are largely independent of layout and library. Read times increase significantly as the number of PEs is increased.**

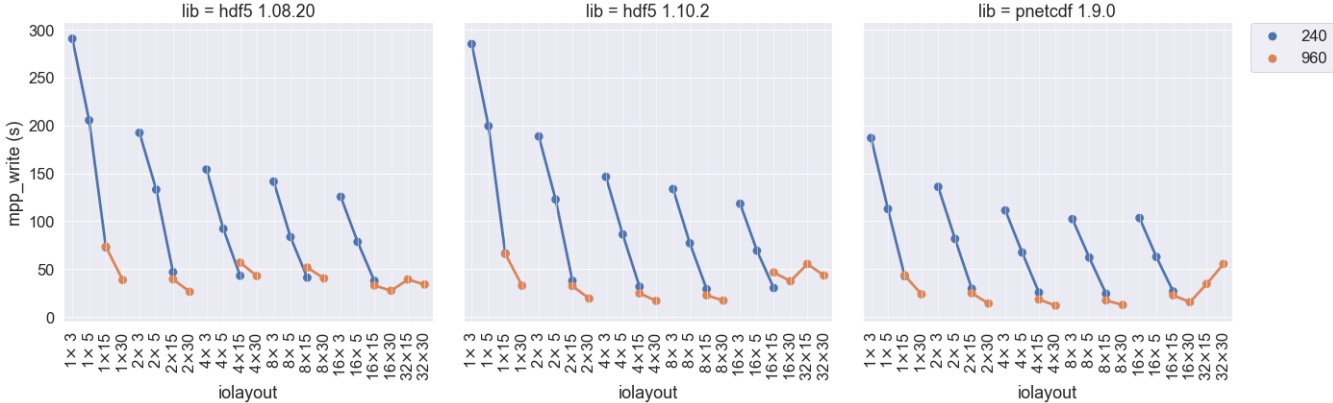

**Figure 5. `mpp_write` time versus I/O layout for different library and formats. Runtimes improve greatly as I/O nodes are** 590 **increased (grouped curves), and modestly as the I/O PEs per node are increased (across grouped curves). Runtimes are scalably reduced as PEs are increased. PnetCDF shows modest improvement over HDF5 performance.**





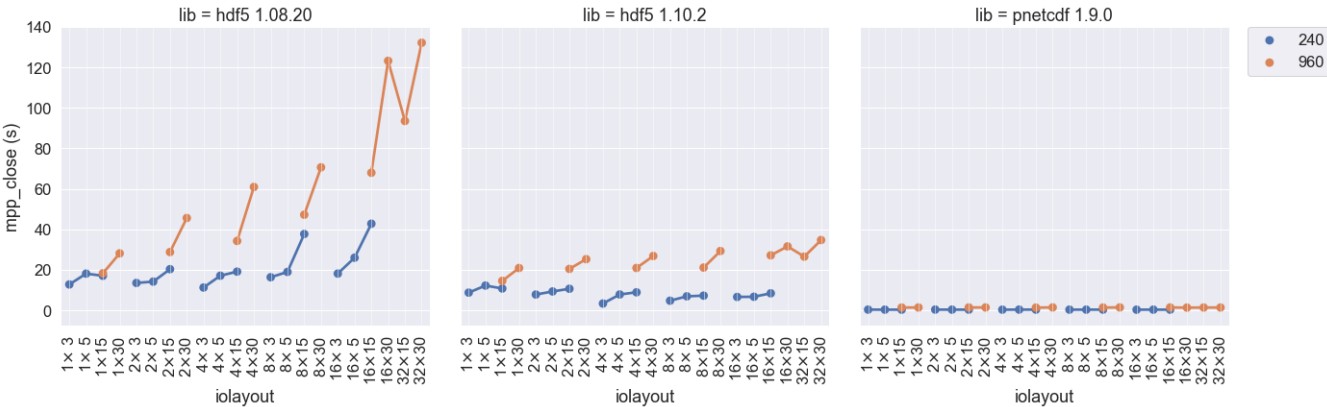

**Figure 6. `mpp_close` time versus I/O layout with different libraries and formats. Contentions within the HDF5 library lead to performance problems, which increase with layout and number of PEs. PnetCDF does not exhibit these issues and close times are negligible.**

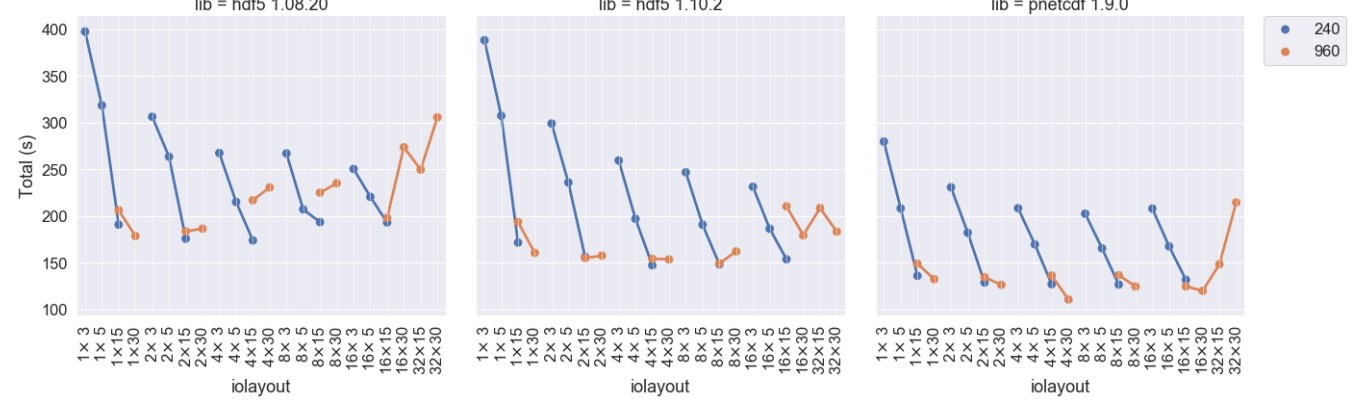


**Figure 7. Total elapsed time versus I/O layout for different libraries and formats. Higher contention at 960 PEs can overwhelm**
**the overall performance trends observed at 240 PEs.**




**Figure 8. The I/O performance of 240-PE benchmarks with different library/format bindings regarding to the number of**
**aggregator and stripe count.**






**Figure 9.** The I/O performance of different library/format bindings with a variety of aggregators and stripe counts by using 960 PEs.






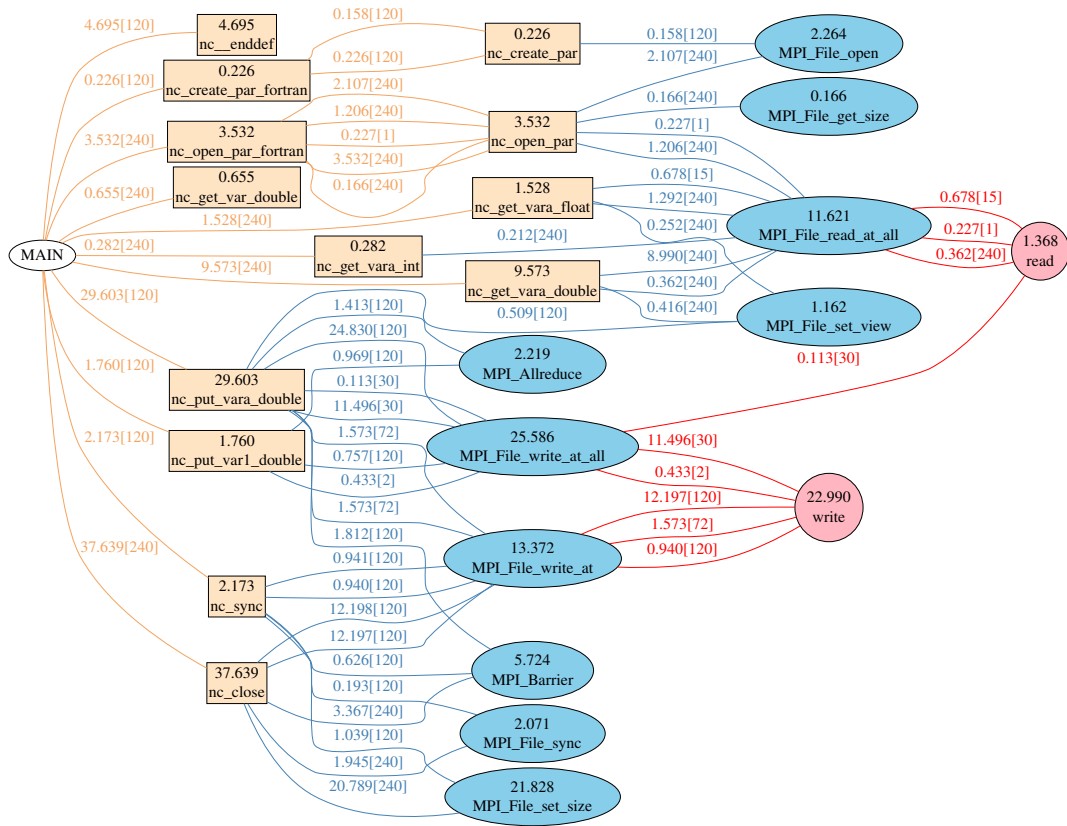

**Figure 10. The call path flow of a tuned 240-PE benchmark with HDF5 1.8.20/netCDF-4. It is classified into 3 layers i.e. netCDF, MPI-IO and system I/O functions. The maximum PE time together with the total number of PEs from the invoker are labelled 630 above each path line and the maximum PE time on each function are labelled within the node block.**



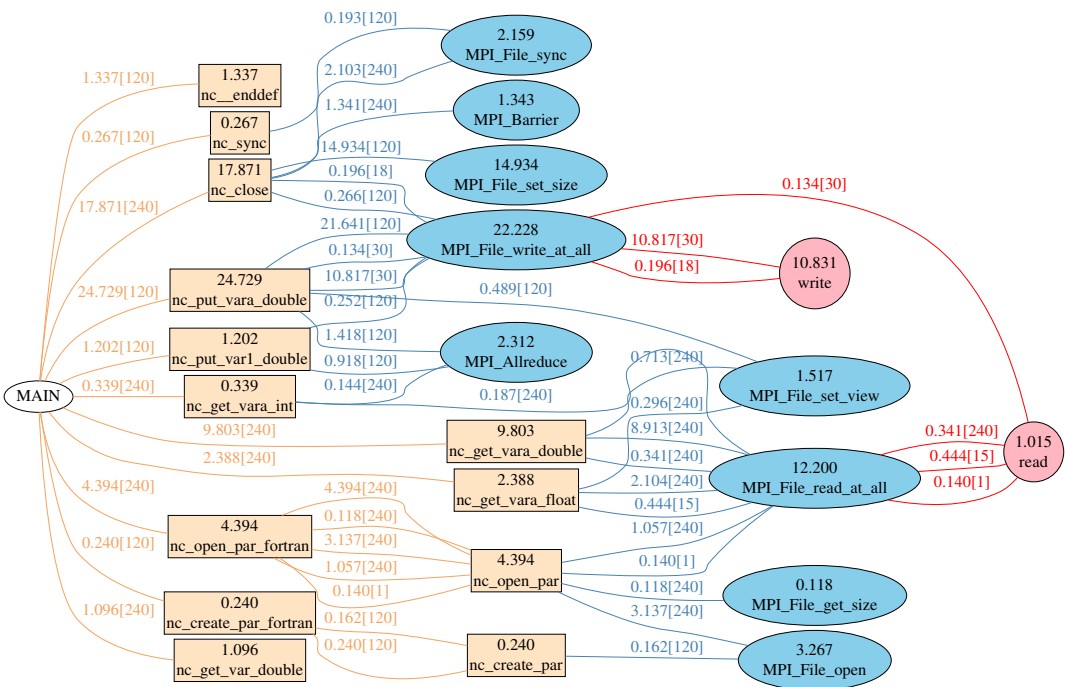

**Figure 11. The call path flow of tuned 240-PE benchmark with HDF5 1.10.2/netCDF-4.**






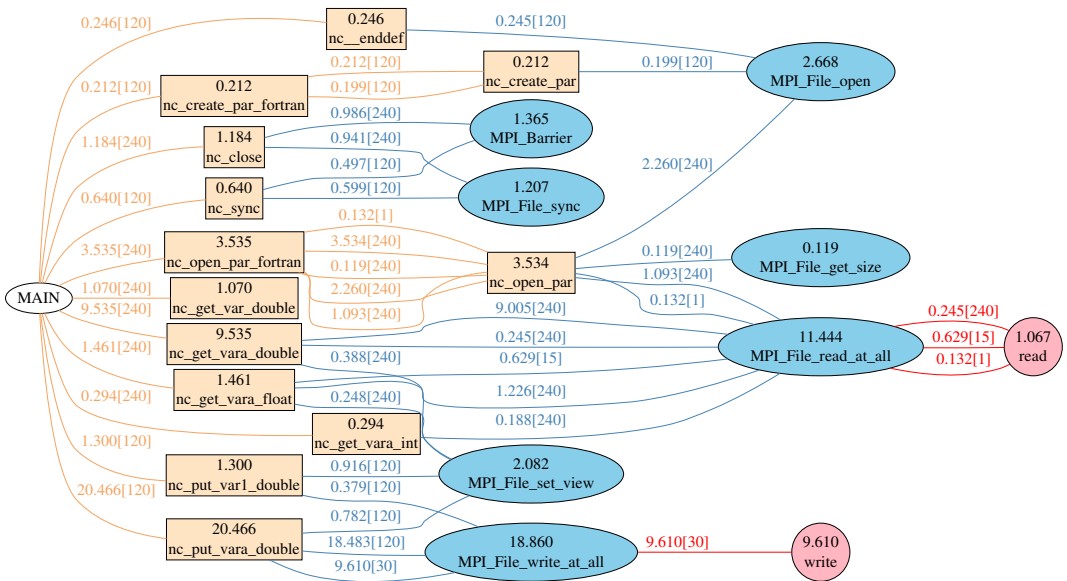

**Figure 12. The callpath flow of tuned 240-PE benchmark with PnetCDF.**

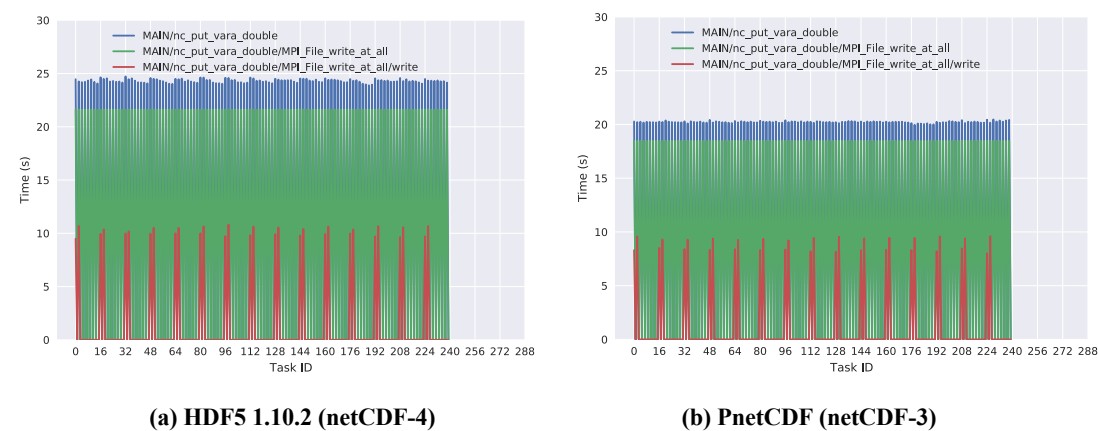

(a) HDF5 1.10.2 (netCDF-4)    (b) PnetCDF (netCDF-3)

**Figure 13. Time distribution over PEs of major write call path functions, i.e. `nc_put_vara_double` for netCDF, `MPI_File_write_at_all` for MPI-IO and POSIX call `write`. The benchmark is running on 240 ranks with an I/O layout of 8✕15.**



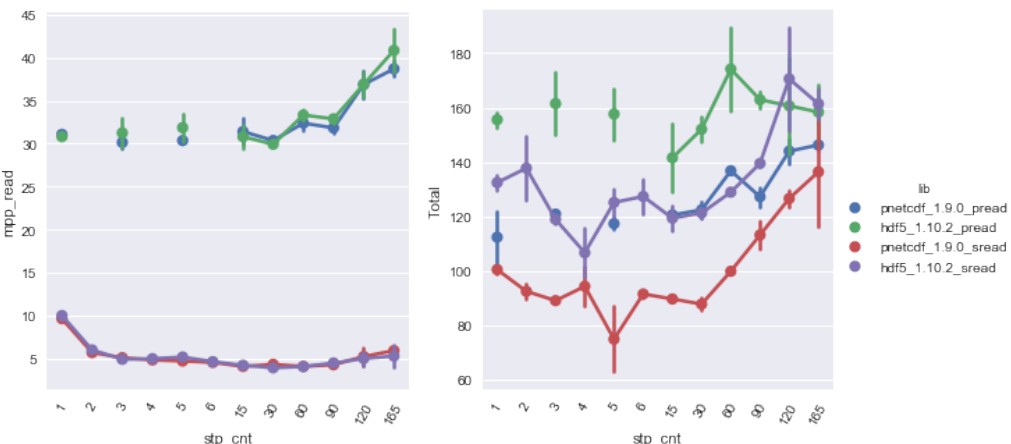

**Figure 14. The 960-PE benchmarks with an I/O layout of 4✕30 and using serial read (`sread`) and parallel read (`pread`) with the HDF5 1.10.2 and PnetCDF libraries. Serial read times are overall more efficient over a range of stripe counts.**

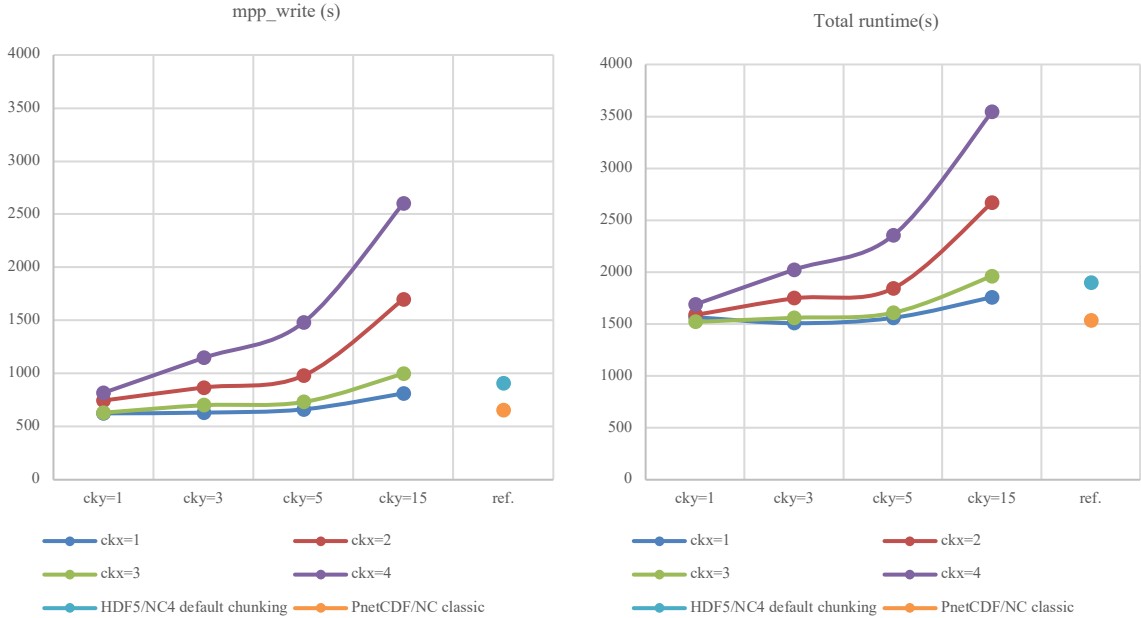

**Figure 15. Performance of 720-PE runs with customized chunk layouts in HDF5/netCDF-4. The default chunk layout of HDF5/netCDF-4 and contiguous layout of PnetCDF/netCDF-3 are shown as references.**




**Table 1. Comparison of write pattern between serial I/O and parallel I/O.**

| Write Pattern | Number of Output File | Run Time | Post-processing Time |
|---|---|---|---|
| Single-threaded, Single File | 1 | Long | None |
| Distributed I/O, Single File per I/O Domain | I/O domains | Moderate | Long |
| Distributed I/O, Single File per PE | PEs | Short | Long |
| Parallel I/O, Single Shared File | 1 | Scalable | None |


**Table 2. The pre-selected parameters at all layers of I/O software stack.**

| Layer | Parameter | Value |
|---|---|---|
| Application | `io_layout` (`iox` × `ioy`) | `iox` = 32, 16, 8, 4, 2, 1 `ioy` = 30,15,5,3 |
| High-level I/O library | Data storage layout | netCDF-3: contiguous netCDF-4: default chunking |
| MPI-IO | `cb_buffer_size` | 64kB |
| | `cb_nodes` | number of PEs |
| | `cb_config_list` | 1, 2, 4, 8 |
| Lustre | `striping_unit` | 1MB |
| | `striping_factor` | 15, 30, 60, 120, 165(max) |






**Table 3. The parallel I/O performance benchmark configurations.**

| Parameters | | Description |
|---|---|---|
| Model | Configurations | 1-day simulations with diagnostic output enabled. 0.25° model (1440×1080) for I/O performance tuning 0.1° model (3600×2700) for validating I/O performance |
| Output | Diagnostic | Diagnostic fields: T, S, $u$, $v$, $t_{age}$ Diagnostic file write frequency: 30 minutes interval for 0.25°, 48 steps, 70GB 5 minutes interval for 0.1°, 288 steps, 2.7TB |
| Benchmark | PEs | 240, 960 for 0.25° model, 720,1440 for 0.1° model |
| | Domain Layout | 16×15 for 240 PEs, 32×30 for 960 PEs (0.25°) 48×15 for 720 PEs, 48×30 for 1440 PEs (0.1°) |
| | I/O library / Format | NetCDF v4.6.1 with the following library/format: HDF5 v1.8.20 / netCDF-4 HDF5 v1.10.2 / netCDF-4, netCDF-4 classic PnetCDF v1.9.0 / netCDF-3 (64-bit offsets) |

**Table 4. Serial single-file I/O time in MOM5 by using 240 and 960 PEs.**

| 0.25° Model | 240 PEs | | 960 PEs | |
|---|---|---|---|---|
| Time (sec.) | netCDF-3 (PnetCDF 1.9.0) | netCDF-4 (HDF5 1.10.2) | netCDF-3 (PnetCDF 1.9.0) | netCDF-4 (HDF5 1.10.2) |
| Total runtime | 637.82 | 687.20 | 629.33 | 671.95 |
| mpp_open | 7.46 | 6.39 | 15.62 | 14.97 |
| mpp_read_meta | 3.90 | 3.73 | 6.16 | 4.88 |
| mpp_read | 4.58 | 4.15 | 2.37 | 2.43 |
| mpp_write | 545.50 | 592.39 | 576.92 | 616.35 |
| mpp_close | 0.65 | 0.96 | 1.23 | 2.37 |




**Table 5. The time metrics of 0.1° model in 720-PE and 1440-PE runs with HDF5 1.10.2/netCDF-4 and PnetCDF 1.9.0/netCDF-3. SIO represents the original serial read and single threaded write; PIO represents the serial read and parallel shared write.**

| Library/Format | HDF5 1.10.2 (netCDF-4) | | | | PnetCDF 1.9.0 (netCDF-3) | | | |
|---|---|---|---|---|---|---|---|---|
| PEs | 720 | | 1440 | | 720 | | 1440 | |
| IO Pattern | SIO | PIO | SIO | PIO | SIO | PIO | SIO | PIO |
| Total runtime (sec.) | 21689 | 1624 | >18000 | 889 | 19726 | 1387 | >18000 | 782 |
| mpp_open (sec.) | 8 | 51 | | 90 | 9 | 16 | | 81 |
| mpp_read (sec.) | 25 | 11 | | 11 | 15 | 11 | | 14 |
| mpp_write (sec.) | 20826 | 705 | | 349 | 18839 | 526 | | 290 |
| mpp_close (sec.) | 8 | 37 | | 59 | 0 | 0 | | 1 |
| Non-I/O Time (sec.) | 828 | 820 | | 380 | 860 | 834 | | 396 |
