# Peer review of "Parallel I/O in FMS and MOM5"

_Geoscientific Model Development, 2019_

## Referee Comment (RC1) · Nikolay V. Koldunov (Referee) · 13 Nov 2019

The paper describes the implementation and tuning of the parallel I/O in MOM.

This is a very good and very timely study, as the parallel I/O continues to be one of the major problems in the current Earth System Science codes. This type of description is usually ether never put together as a text or in a best-case scenario just "collect dust" as a technical report. The authors did a great job of describing in detail their technical development as a paper, and I wish there are more such descriptions in the future. The sharing of this information is very important so that the progress in the field is faster.

I am in general happy with the paper, while having a couple of suggestions that authors free to agree or disagree with. Paper can be published after minor revision.

General points

To make the paper even more useful it would be nice to discuss several additional details.

Short description of how hard it was to implement parallel I/O using each of the libraries (maybe person/month estimate?), what is the user experience with each of the libraries (are they easy to install and support?).

Mentioning another parallel I/O solutions, that become popular in Earth Science (e.g. XIOS http://www.ifremer.fr/docmars/html/doc.coupling.xios.html ) or even something outside of the ocean modelling world (e.g. https://csmd.ornl.gov/adios ) would help the unexperienced reader to be more aware of available software solutions.

Maybe you can speculate about the applicability of your results to unstructured mesh ocean models, that usually store their results in netCDF as long 1D vectors?

Your data-intensive benchmark, although it serves the purpose well, is not very realistic. I think your results will shine even more if you can show how beneficial parallel I/O is in realistic simulations. In Koldunov et al., 2019 we showed that in our case for relatively small setup (about 600 000 surface points) running on 1152 cores the price of the serial I/O in "operational" simulation is only about 5%. For the user that typically has tasks of this size it is not a very large price to pay, and maybe investments in the parallel I/O are not necessary. It would be great if you can run, say, a year of model simulations with typical I/O workload (e.g. in our case its monthly means) on different number of cores with serial and parallel I/O and estimate the amount of time (in %) the I/O takes from the total run time.

Minor point

For figures 3 to 7 please add PE/node as a second x-axis (e.g. on the top). This will make it easier to interpret.

Nikolay Koldunov

---

## Author Comment (AC1) · 16 Jan 2020

Referee's Comments: The paper describes the implementation and tuning of the parallel I/O in MOM. This is a very good and very timely study, as the parallel I/O continues to be one of the major problems in the current Earth System Science codes. This type of description is usually ether never put together as a text or in a best-case scenario just "collect dust" as a technical report. The authors did a great job of describing in detail their technical development as a paper, and I wish there are more such descriptions in the future. The sharing of this information is very important so that the progress in the field is faster.

Response: We appreciate referee's comments as above and agree that it is important to share experience and skills in enhancing I/O performance of the climate and earth system models.

Referee's Comments: I am in general happy with the paper, while having a couple of suggestions that authors free to agree or disagree with. Paper can be published after minor revision. General points To make the paper even more useful it would be nice to discuss several additional details. Short description of how hard it was to implement parallel I/O using each of the libraries (maybe person/month estimate?), what is the user experience with each of the libraries (are they easy to install and support?).

Response: FMS has provided enough functionality to allow us to use the netCDF library directly (please see response to the next comment) while the parallel I/O to netCDF-4 and classic files are achieved through the HDF5 and PnetCDF libraries respectively. All these libraries are widely used in scientific computing; they follow standard installation process (e.g. autotools) and are well supported. It is straightforward to implement I/O operations in the code by invoking the netCDF library APIs after some extra work to set up the I/O domain communicator. The following sentences have been added to the revised manuscript to estimate the time of development work: "Development required approximately one month to implement a working feature, along with an additional month of work to troubleshoot more complex configurations related to land masking and the handling of I/O domains which only cover a subset of the total grid." We also spent plenty of time and effort on the I/O performance tuning, as there are lots of possible bindings among parameters from multiple I/O layers spanning MOM5 I/O domain, netCDF library, MPI-IO and lustre file system. It proves that these efforts are necessary to achieve the optimized parallel I/O performance.

Referee's Comments: Mentioning another parallel I/O solutions, that become popular in Earth Science (e.g. XIOS http://www.ifremer.fr/docmars/html/doc.coupling.xios.html ) or even something outside of the ocean modelling world (e.g. https://csmd.ornl.gov/adios ) would help the unexperienced reader to be more aware of available software solutions.

Response: XIOS is an attractive I/O framework with the capability to provide highly scalable I/O performance and it has been used in many climate and earth models. We

excluded XIOS as the I/O solution in this work because there is a need to maintain the current I/O pattern of FMS in which compute PEs take part in I/O activities rather than setting up the dedicated I/O server with extra PEs as XIOS does. However, the possibility on implementing XIOS in future version is always open. We also ruled out ADIOS as the candidate solution, although it appears to be highly scalable, as it doesn't directly support NetCDF and it requires conversion to NetCDF. It has not yet been used in the weather and climate domain. We have added the following paragraph to Section 2 explaining why FMS was sufficient for our approach. "Because FMS provides access to distributed datasets as well as a mechanism for collecting the data into larger I/O domains for writing to disk, we concluded that FMS already contained much of the functionality provided by existing parallel I/O libraries, and that it would be more efficient to generalize the I/O domain for both writing to files and passing data to a general-purpose IO libraries such as netCDF. In this sense, there is no need to set up the dedicated I/O server with extra PEs as other popular parallel I/O solution like XIOS does."

Referee's Comments: Maybe you can speculate about the applicability of your results to unstructured mesh ocean models, that usually store their results in netCDF as long 1D vectors?

Response: Although we are not very familiar with the implementation details of unstructured models, we believe that there may be a benefit to using the methods detailed here if the data for each mesh element are stored contiguously, and the buffers do not need to be populated from complex data structures associated with the mesh. We have added the following paragraph to the discussion commenting on this issue: "Although this work is applied to a model with a fixed regular grid, these results could be applied to a model with an unstructured mesh. Much of the work required to populate the I/O domains and to define chunked regions is required to produce contiguous streams of data which are passed to the I/O library. If the data is already stored as contiguous 1D arrays, then the task of dividing the data across I/O servers could be trivial. If more

complex data structures are used, such as linked lists, then the buffering of data into contiguous arrays could add significant overhead to parallel I/O."

Referee's Comments: Your data-intensive benchmark, although it serves the purpose well, is not very realistic. I think your results will shine even more if you can show how beneficial parallel I/O is in realistic simulations. In Koldunov et al., 2019 we showed that in our case for relatively small setup (about 600 000 surface points) running on 1152 cores the price of the serial I/O in "operational" simulation is only about 5%. For the user that typically has tasks of this size it is not a very large price to pay, and maybe investments in the parallel I/O are not necessary. It would be great if you can run, say, a year of model simulations with typical I/O workload (e.g. in our case its monthly means) on different number of cores with serial and parallel I/O and estimate the amount of time (in %) the I/O takes from the total run time.

Response: We agree with the reviewer that it would be valuable to consider the potential benefits of parallel I/O in more realistic simulations, and have included results from 8-day simulations with 1-day and 4-day I/O frequencies in a new table (Table 6). The serial I/O takes around 6% of total runtime in 720-PE runs which could be regarded as typical I/O workload. The benchmark results indicate that parallel I/O can reduce the I/O time ratio to be less than 1%. More importantly, the serial I/O time ratio of the 1440-PE simulation is about 11%, indicating that the serial I/O may eventually become the parallel performance bottleneck. The parallel write time, on the other hand, scales well with the number of PEs and may prevent I/O from blocking the overall performance scalability.

Referee's Comments: Minor point For figures 3 to 7 please add PE/node as a second x-axis (e.g. on the top). This will make it easier to interpret.

Response: This is a very good suggestion. In this paper the same I/O layout may exist in both 720-PE and 1440-PE simulations. For example, the I/O layout 2*15 could be given by 2 PEs per node*15 nodes in 720-PE or 1 PE per node*30 nodes in 1440-PE

simulations respectively. Thus it is impracticable to make the second x-axis on the top as the same I/O layouts may repeat at the first x-axis. Alternatively, we append the PE distribution i.e. [PE per node * nodes] to I/O layout at the x-axis in those revised Figures (Fig. 3~7) to clarify the connection between I/O layout and PE distribution.

The revised figures 3~7 and Table 6 are put into the supplement document for reviewing.

Please also note the supplement to this comment:
https://www.geosci-model-dev-discuss.net/gmd-2019-257/gmd-2019-257-AC1-supplement.pdf

**Supplement:**

[Figure]

**Figure 3.** `mpp_open` time (in sec.) versus I/O layout in different libraries and PE numbers. HDF5 times are generally larger than in PnetCDF, and the runtime increases as PEs increase from 240 to 960. The I/O layout together with its PE distribution in [PE per node × nodes] are labelled in X-axis.

[Figure]

**Figure 4.** `mpp_read` time (in sec.) versus I/O layout in different libraries and PE numbers. Read operations do not use I/O layout or parallel I/O, and runtimes are largely independent of layout and library. Read times increase significantly as the number of PEs is increased. The I/O layout together with its PE distribution in [PE per node × nodes] are labelled in X-axis.

[Figure]

**Figure 5.** `mpp_write` **time versus I/O layout for different library and PE numbers. Write time improves greatly as I/O nodes are increased (grouped curves), and modestly as the I/O PEs per node are increased (across grouped curves). Runtimes are scalarly reduced as PEs are increased. PnetCDF shows modest improvement over HDF5 performance. The I/O layout together with its PE distribution in [PE per node × nodes] are labelled in X-axis.**

[Figure]

**Figure 6.** `mpp_close` **time versus I/O layout with different libraries and PE numbers. Contentions within the HDF5 library lead to performance problems, which increase with layout and number of PEs. PnetCDF does not exhibit these issues and close times are negligible. The I/O layout together with its PE distribution in [PE per node × nodes] are labelled in X-axis.**

[Figure]

**Figure 7. Total elapsed time versus I/O layout for different libraries and PE numbers. Higher contention at 960 PEs can overwhelm the overall performance trends observed at 240 PEs. The I/O layout together with its PE distribution in [PE per node × nodes] are labelled in X-axis.**

30

**Table 6. The time metrics of 0.1° model in 720-PE and 1440-PE runs with less I/O frequencies, i.e. write per 1 day and 4 days in 8-day simulations. SIO represents the original single threaded write; PIO represents parallel shared write. The I/O time composes of contributions from** `mpp_open`, `mpp_read`, `mpp_write` **and** `mpp_close`**. The I/O time ratio is given between the I/O time and total runtime.**

| I/O pattern&Format | | SIO in netCDF4_classic | | PIO in netCDF-4 | | PIO in netCDF-3 | |
|---|---|---|---|---|---|---|---|
| I/O frequency | | 1-day | 4-day | 1-day | 4-day | 1-day | 4-day |
| 720 PEs | Total runtime (sec.) | 8114 | 7817 | 7685 | 7569 | 7666 | 7469 |
| | I/O time / `mpp_write` (sec.) | 494/453 | 302/265 | 75/40 | 62/27 | 57/17 | 49/11 |
| | I/O ratio | 6.09% | 3.87% | 0.98% | 0.82% | 0.74% | 0.66% |
| 1440 PEs | Total runtime (sec.) | 4118 | 3743 | 3547 | 3578 | 3518 | 3549 |
| | I/O time / `mpp_write` (sec.) | 452/421 | 269/238 | 59/24 | 48/14 | 51/14 | 40/7 |
| | I/O time ratio | 10.98% | 7.18% | 1.67% | 1.35% | 1.45% | 1.14% |

---

## Short Comment (SC1) · 29 Jan 2020

This is an executive editor comment highlighting the ways in which this manuscript is not currently compliant with GMD policy on code and data availability. In this case, there is just a single technical issue which needs to be remedied in the revised submission:

1. Github URL. Github is an excellent development platform, but it lacks the features required of an archive. GitHub themselves tell authors to use Zenodo for this purpose. The authors should follow the procedure detailed here to archive the exact version of the software used to create the results presented: https://guides.github.com/activities/citable-code/ . The resulting Zenodo repositories present the correct bibliography entries to use.

2. No data identified. The datasets used to conduct the evaluation experiments presented must be cited from the code and data availability section with enough precision to allow a reader to reproduce the work in the manuscript.

3. No configuration, run, or data processing scripts. The configuration files, run scripts and any data processing or analysis scripts used to produce the results presented in the manuscript need to be publicly and persistently archived, and cited from the code and data availability section. As a guide, every file the user would need to reproduce the manuscript should accessible.

Further details on code and data availability requirements are in the GMD model code and data policy: https://www.geoscientific-model-development.net/about/code_ and_data_policy.html. The reasons for the policy and more detail are provided in this editorial: https://doi.org/10.5194/gmd-12-2215-2019.

---

## Referee Comment (RC2) · Michael Kuhn (Referee) · 2 Mar 2020

The authors present a detailed study of implementing parallel I/O using NetCDF in the Modular Ocean Model version 5 via the Flexible Modelling System. Even though the implementation is quite specific to MOM5, the paper can serve as a useful experience for developers aiming to implement parallel I/O within other scientific software packages. Overall, I believe the paper is worth publishing, especially since I/O aspects are often neglected. There are still some points for improvement, though.

Specific comments:

- Lines 36-38: Where is the number of 350 MB/s for disk throughput coming from? The HDDs I know about typically max out at roughly 200 MB/s. While I understand the point you are trying to make with these sentences, I believe some more details would make them easier to follow. How long does a one-year simulation typically take? Is writing

out one terabyte of data even relevant in this case?

- Lines 87-96: Please elaborate why you have selected NetCDF for your parallelization efforts. There are also other approaches such as SIONlib or ADIOS. While NetCDF probably makes the most sense for geoscientific applications, this should at least be discussed briefly.

- Lines 191-195: Have you considered the alignment of chunks? We have shown in "A Best Practice Analysis of HDF5 and NetCDF-4 Using Lustre (Bartz, Chasapis, Kuhn, Nerge, Ludwig)" that chunk alignment can have very significant impact on parallel I/O performance. Sadly, NetCDF did not (and apparently still does not) expose this functionality while HDF5 does. It is therefore necessary to patch NetCDF to enable HDF5's chunk alignment. Missing alignment could be the cause of contention you describe when increasing the number of I/O PEs per I/O domain.

- Lines 295-299: See previous comment, this could also be caused by missing alignment.

- Lines 451-453: The serial I/O versions with 720 PEs ran for 6 hours while the ones with 1440 PEs were killed after 5 hours. Did the 720 PE version run on a different partition? If so, is it still possible to compare the two?

- Lines 508-512: Why did you develop your own I/O profiling tool? There are existing options such as Score-P or Darshan. Please state why the existing tools did not meet your requirements.

- Line 526: I gave the GitHub repository a quick look but could only find the source code. According to GMD's code and data policy, the data must also be provided. You have also not mentioned in the paper which commit you were using to perform the model runs.

Technical corrections:

- Line 28: The acronym OS has been introduced before in line 23 and does not need

to be repeated here.

- Lines 62-70: Since you talk about "single file I/O" in the paragraph before, it might be worth mentioning explicitly that one file is created per I/O domain in this case.

- Line 73: "A typical 0.25° global simulations ..." - It should be "simulation".

- Line 183: "... in Table 3, ..." - This should be "Table 2".

- Line 227: "... of the I/O parameters in Table 3." - Should be "Table 2".

- Line 238: "... grids are disturbed over ..." - This should probably be "distributed".

- Line 375: "... in the charts below for each library." - This should rather reference the figures directly since they are placed in the appendix.

- Line 429: "... in Figure 14." - Figure 14 seems to be rather blurry while the others are fine. Please provide a high-resolution version if possible.

- Lines 581-622: Are the reported values averages? If so, you should mention this somewhere and also give deviations. Figure 14 already includes them but the others do not.

- Lines 625-639: Bright orange is hard to read on white, so it might make sense to change the color for the profiling graphs.

- Line 665: "Number of Output File" - This should be "Files".

- Lines 680-685: To better assess the scaling behavior, please also mention the number of nodes in addition to the number of PEs.

---

## Author Comment (AC2) · 15 Mar 2020

We have updated the 'Code availability' section of the manuscript to include more details as below:

The source code of parallel I/O enabled FMS is available from doi:org/10.5281/zenodo.3700099. The MOM5 code used in the work is available at https://github.com/mom-ocean/MOM5.git. The core dataset is available as doi:10.1007/s00382-008-0441-3. Build script, configure files and job scripts are available from dio:org/10.5281/zenodo.3710732.

---

## Author Comment (AC3) · 15 Mar 2020

Referee's Comments: The authors present a detailed study of implementing parallel I/O using NetCDF in the Modular Ocean Model version 5 via the Flexible Modelling System. Even though the implementation is quite specific to MOM5, the paper can serve as a useful experience for developers aiming to implement parallel I/O within other scientific software packages. Overall, I believe the paper is worth publishing, especially since I/O aspects are often neglected. There are still some points for improvement, though.

Specific comments:

Referee's Comments:

- Lines 36-38: Where is the number of 350 MB/s for disk throughput coming from? The HDDs I know about typically max out at roughly 200 MB/s. While I understand the point

you are trying to make with these sentences, I believe some more details would make them easier to follow.

Response:

The 350 MB/s performance was based on measurements of direct disk writes (using 'dd') for idealized output on the machine (Raijin), and was the performance typically reported in most technical specifications of the machine. Although this machine has since been decommissioned, 350 MB/s is the cited performance of the "gdata1" filesystem; see the following presentation by Daniel Rodwell from 2016, slide 27 (https://www.eofs.eu/_media/events/lad16/05_petascale_data_migration_rodwell.pdf).

We also note that consumer SATA SSD speeds of 500 MB/s are not uncommon, and the presentation above cites Lustre OST write speeds as high as 800 MB/s. So our estimate of 350 GB/s seems to be reasonable for the purpose of discussion at this early stage of the paper. Given the large variation in write speed performance, we do not have a good reference but are welcome to suggestions from the reviewer.

Referee's Comments:

How long does a one-year simulation typically take? Is writing out one terabyte of data even relevant in this case?

Response:

We believe that the sentences preceding the discussion of terabyte-per-year output justify this output rate. A typical $0.1°$ grid (3600 x 2700) with 75 levels at double precision will require approx. 5.8 GB per step. At a 5-day output rate, this will require over 400 GB, and the output could have many such fields.

There is no simple way to characterize a typical model output rate, but we believe that the information above justifies that even a minimal high-resolution experiment will produce output on the order of terabytes per year.

As for whether a year represents a typical climate simulation time, we felt that this needed no citation.

A typical runtime of a high resolution model would be on the order of 10 hours per year. But the compute runtime or the ratio of compute to I/O time does not change the fact that terabytes of data must be written, and the preceding sentences establish that it is a reasonable workload for a high resolution ocean model. It must be done, and it would require hours of time to complete if it were done serially.

Our purpose was to demonstrate that serial I/O of a high resolution model is a prohibitive task, and we feel that the leading statements justify this statement. But if the reviewer disagrees with any of the statements above, we are happy to address them.

Referee's Comments:

- Lines 87-96: Please elaborate why you have selected NetCDF for your parallelization efforts. There are also other approaches such as SIONlib or ADIOS. While NetCDF probably makes the most sense for geoscientific applications, this should at least be discussed briefly.

Response:

In a later revision of the paper as a response to the first referee, we explain that I/O domain of FMS provides most of the functionality of these libraries, and therefore opted to directly implement parallel I/O based on the existing I/O domain structure. The following discussions have been added in Section 2 of the revised manuscript:

"Because FMS provides access to distributed datasets as well as a mechanism for collecting the data into larger I/O domains for writing to disk, we concluded that FMS already contained much of the functionality provided by existing parallel I/O libraries, and that it would be more efficient to generalize the I/O domain for both writing to files and passing data to a general-purpose IO libraries such as netCDF. In this sense, there is no need to set up the dedicated I/O server with extra PEs as other popular parallel

I/O solution like XIOS does."

Referee's Comments:

- Lines 191-195: Have you considered the alignment of chunks? We have shown in "A Best Practice Analysis of HDF5 and NetCDF-4 Using Lustre (Bartz, Chasapis, Kuhn, Nerge, Ludwig)" that chunk alignment can have very significant impact on parallel I/O performance. Sadly, NetCDF did not (and apparently still does not) expose this functionality while HDF5 does. It is therefore necessary to patch NetCDF to enable HDF5's chunk alignment. Missing alignment could be the cause of contention you describe when increasing the number of I/O PEs per I/O domain.

- Lines 295-299: See previous comment, this could also be caused by missing alignment.

Response:

In this paper we focus on the configurable parameters associated with MOM5 I/O domain layout, the netCDF library (based on standard HDF5 installation), MPI-IO, and the Lustre file system. The impact of chunk alignment configurable by HDF5 is an interesting idea worthy of further exploration, and it may help to explain some of the performance differences between PnetCDF and HDF5, but we feel that it is perhaps beyond the scope of this paper, i.e. it is not tuneable via netCDF library.

Referee's Comments:

- Lines 451-453: The serial I/O versions with 720 PEs ran for 6 hours while the ones with 1440 PEs were killed after 5 hours. Did the 720 PE version run on a different partition? If so, is it still possible to compare the two?

Response:

Both 720 PE and 1440 PE jobs run on the same partition, but the latter has a shorter time limit, i.e. 5 hours, set by the PBS queue system. At this stage, it is not possible

to compare the two as the machine has been decommissioned. We can only present it as an incompletable task on our platform, which we believe is sufficient for the more detailed analysis of the parallel I/O performance. However, as per request of the first referee, we added a new table (Table 6 in the revised manuscript) to compare simulations with typical I/O loads. In that table, both serial I/O and parallel I/O are compared between 720 PEs and 1440 PEs but with much less I/O loads than those in Table 5.

Referee's Comments:

- Lines 508-512: Why did you develop your own I/O profiling tool? There are existing options such as Score-P or Darshan. Please state why the existing tools did not meet your requirements.

Response:

We would like to analyse costs of each site in major I/O call paths by collecting the elapsed time and sizes for all MPI ranks at multiple I/O layers such as NetCDF, MPI-IO and POSIX I/O calls. The existing I/O profilers, however, cannot fully approach this goal. Score-P is good at profiling user code and MPI-IO functions, but it is hard to measure the time spent within the netCDF library and POSIX calls. Also, it cannot measure the elapsed time and size per I/O operation for each individual input and output file. Darshan, on the other hand, is good at profiling the time and size of I/O operations of different files. However, it cannot provide the rank distribution of time which is necessary to analyse the load balance issue.

By recognizing the above deficiencies of existing I/O profilers, we decided to develop our own profile tool which can address above issues with negligible overheads. The tool can provide all details we need to evaluate the cost of each I/O layer, rank distribution of time per file, access size per I/O function or operation and so on.

Referee's Comments:

- Line 526: I gave the GitHub repository a quick look but could only find the source

code. According to GMD's code and data policy, the data must also be provided. You have also not mentioned in the paper which commit you were using to perform the model runs.

Response:

These issues were also raised by the executive editor. We have updated more details about the code and data availability as below:

The source code of parallel I/O enabled FMS is available from doi:org/10.5281/zenodo.3700099. The MOM5 code used in the work is available at https://github.com/mom-ocean/MOM5.git. The core dataset is available as doi:10.1007/s00382-008-0441-3. Build script, configure files and job scripts are available from dio:org/10.5281/zenodo.3710732.

Referee's Comments:

Technical corrections: - Line 28: The acronym OS has been introduced before in line 23 and does not need to be repeated here.

Response: Removed acronym OS.

Referee's Comments:

- Lines 62-70: Since you talk about "single file I/O" in the paragraph before, it might be worth mentioning explicitly that one file is created per I/O domain in this case.

Response: Explicitly cite 4 write patterns of Table 1 in the context.

Referee's Comments:

- Line 73: "A typical 0.25âŮę global simulations ..." - It should be "simulation".

Response: Fixed.

Referee's Comments:

- Line 183: "... in Table 3, ..." - This should be "Table 2".

Response: Fixed.

Referee's Comments:

- Line 227: "... of the I/O parameters in Table 3." - Should be "Table 2".

Response: Fixed.

Referee's Comments:

- Line 238: "... grids are disturbed over ..." - This should probably be "distributed".

Response: Fixed.

Referee's Comments:

- Line 375: "... in the charts below for each library." - This should rather reference the figures directly since they are placed in the appendix.

Response: Reference the figures directly.

Referee's Comments:

- Line 429: "... in Figure 14." - Figure 14 seems to be rather blurry while the others are fine. Please provide a high-resolution version if possible.

Response: Figure 14 is reproduced with the higher resolution.

Referee's Comments:

- Lines 581-622: Are the reported values averages? If so, you should mention this somewhere and also give deviations. Figure 14 already includes them but the others do not.

Response: Clarified in figure caption that the reported values are maximum among all PEs.

Referee's Comments:

- Lines 625-639: Bright orange is hard to read on white, so it might make sense to change the color for the profiling graphs.

Response: The light color has been replaced by the deeper one.

- Line 665: "Number of Output File" - This should be "Files".

Response: Fixed.

Referee's Comments:

- Lines 680-685: To better assess the scaling behavior, please also mention the number of nodes in addition to the number of PEs.

Response: Added node counts in Table 5.

Please also note the supplement to this comment:
https://www.geosci-model-dev-discuss.net/gmd-2019-257/gmd-2019-257-AC3-supplement.pdf

**Supplement:**

[Figure]

**Figure 10. The call path flow of a tuned 240-PE benchmark with HDF5 1.8.20/netCDF-4. It is classified into 3 layers i.e. netCDF, MPI-IO and system I/O functions. The maximum PE time together with the total number of PEs from the invoker are labelled above each path line and the maximum PE time on each function are labelled within the node block.**

[Figure]

**Figure 11. The call path flow of tuned 240-PE benchmark with HDF5 1.10.2/netCDF-4. It is classified into 3 layers i.e. netCDF, MPI-IO and system I/O functions. The maximum PE time together with the total number of PEs from the invoker are labelled above each path line and the maximum PE time on each function are labelled within the node block.**

[Figure]

**Figure 12. The callpath flow of tuned 240-PE benchmark with PnetCDF. It is classified into 3 layers i.e. netCDF, MPI-IO and system I/O functions. The maximum PE time together with the total number of PEs from the invoker are labelled above each path line and the maximum PE time on each function are labelled within the node block.**

**Table 6. The time metrics of 0.1° model in 720-PE and 1440-PE runs with less I/O frequencies, i.e. write per 1 day and 4 days in 8-day simulations. SIO represents the original single threaded write; PIO represents parallel shared write. The I/O time composes of contributions from `mpp_open`, `mpp_read`, `mpp_write` and `mpp_close`. The I/O time ratio is given between the I/O time and total runtime. All values are taken from the maximum walltime among all PEs.**

| I/O pattern&Format | | SIO in netCDF4_classic | | PIO in netCDF-4 | | PIO in netCDF-3 | |
|---|---|---|---|---|---|---|---|
| I/O frequency | | 1-day | 4-day | 1-day | 4-day | 1-day | 4-day |
| 720 PEs | Total runtime (sec.) | 8114 | 7817 | 7685 | 7569 | 7666 | 7469 |
| | I/O time / `mpp_write` (sec.) | 494/453 | 302/265 | 75/40 | 62/27 | 57/17 | 49/11 |
| | I/O ratio | 6.09% | 3.87% | 0.98% | 0.82% | 0.74% | 0.66% |
| 1440 PEs | Total runtime (sec.) | 4118 | 3743 | 3547 | 3578 | 3518 | 3549 |
| | I/O time / `mpp_write` (sec.) | 452/421 | 269/238 | 59/24 | 48/14 | 51/14 | 40/7 |
| | I/O time ratio | 10.98% | 7.18% | 1.67% | 1.35% | 1.45% | 1.14% |